# Retention time prediction for chromatographic enantioseparation by quantile geometry-enhanced graph neural network

Hao Xu [1,2], Jinglong Lin[1], Dongxiao Zhang [3,4] ✉ & Fanyang Mo [1,5] ✉

The enantioseparation of chiral molecules is a crucial and challenging task in the field of experimental chemistry, often requiring extensive trial and error with different experimental settings. To overcome this challenge, here we show a research framework that employs machine learning techniques to predict retention times of enantiomers and facilitate chromatographic enantioseparation. A documentary dataset of chiral molecular retention times in high-performance liquid chromatography (CMRT dataset) is established to handle the challenge of data acquisition. A quantile geometry-enhanced graph neural network is proposed to learn the molecular structure-retention time relationship, which shows a satisfactory predictive ability for enantiomers. The domain knowledge of chromatography is incorporated into the machine learning model to achieve multi-column prediction, which paves the way for chromatographic enantioseparation prediction by calculating the separation probability. The proposed research framework works well in retention time prediction and chromatographic enantioseparation facilitation, which sheds light on the application of machine learning techniques to the experimental scene and improves the efficiency of experimenters to speed up scientific discovery.

In recent years, the rapid development of machine learning intelligence has brought prosperity to the field of 'machine learning for chemistry'[1], which spawns a series of applications including molecular properties prediction[2], drug discovery[3], and retrosynthetic analysis[4–6]. Although diversified machine learning models have been invented to accomplish requirements in many research scenarios[7–9], fundamental limitations still lie in the aspects of dataset generation and molecular representations, which hinder the integration of machine learning and chemistry.

Datasets are fundamental to machine learning since the quantity and quality of data directly relate to the performance of machine learning models. Unfortunately, the generation of chemical data is usually time-consuming and labor-intensive due to the experimental attributes in chemistry. Therefore, high-throughput techniques combined with automation have been developed to accumulate standardized experimental data efficiently[10,11]. For example, our prior work[12] has created an automated platform to conduct high-throughput thin-layer chromatographic analysis. However, high-throughput systems are usually expensive and targeted at specific scenarios, which is difficult to promote to broader fields. An alternative way is collecting data from published articles, but the quality usually varies from objective

[1]School of Materials Science and Engineering, Peking University, 100871 Beijing, P. R. China. [2]BIC-ESAT, ERE, and SKLTCS, College of Engineering, Peking University, 100871 Beijing, P. R. China. [3]Eastern Institute for Advanced Study, Eastern Institute of Technology, 315200 Ningbo, Zhejiang, P. R. China. [4]Department of Mathematics and Theories, Peng Cheng Laboratory, 518000 Shenzhen, Guangdong, P. R. China. [5]AI for Science (AI4S)-Preferred Program, Peking University Shenzhen Graduate School, 518055 Shenzhen, P. R. China. ✉e-mail: dzhang@eias.ac.cn; fmo@pku.edu.cn

factors. It means that the uncertainty of data needs to be taken into consideration.

Molecular representation is another issue that needs to be handled properly. Chemical molecules usually have a variety of classic representation ways, including SMILES[13], fingerprints[14], and descriptors[15]. Although these ways have achieved gratifying performance in constructing quantitative structure-activity relationships (QSAR), they have difficulty in representing 3D conformer-related properties like chirality (Fig. 1a), which confines their further application. Fortunately, the derivatives of the graph neural network (GNN), including geometry-enhanced graph neural network (GeoGNN)[16] and Uni-mol[17], attempted to incorporate 3D information to enhance the molecular graph representation. However, massive data are required for training, which is unaffordable in the experimental scene where data are usually scarce and expensive.

In view of the above-mentioned pain points, we explore a research framework to incorporate machine learning techniques into practical problems in experimental chemistry. The prediction for chromatographic enantioseparation is presented as a persuasive example in this work, which is of great significance in synthetic chemistry, material science, and biopharmaceutical[18–20]. High-performance liquid chromatography (HPLC) is the mainstream way for chromatographic enantioseparation[21]; however, the choice of the experimental condition needs trial-and-error, which is tedious and time-consuming since each trial may take tens of minutes (Fig. 1b). Therefore, in this work, we construct a machine learning model that can predict the retention time (RT) of given chiral molecules and recommend the most suitable condition with the highest possibility of separation (Fig. 1b). Different from previous works that focus merely on the quantitative structure–retention relationship (QSRR) models[22–25], we take a step further to consider how the machine learning models can promote chemical experiments practically. To this end, we pay more attention to the normal-phase HPLC that is usually employed to separate chiral molecules instead of reversed-phase HPLC in existing literatures[25,26]. Our contribution can be summarized as follows:

1. A chiral molecular retention time dataset (CMRT dataset) is established in this work by collecting experimental data reported in 644 articles about asymmetric catalysis. The CMRT dataset constitutes

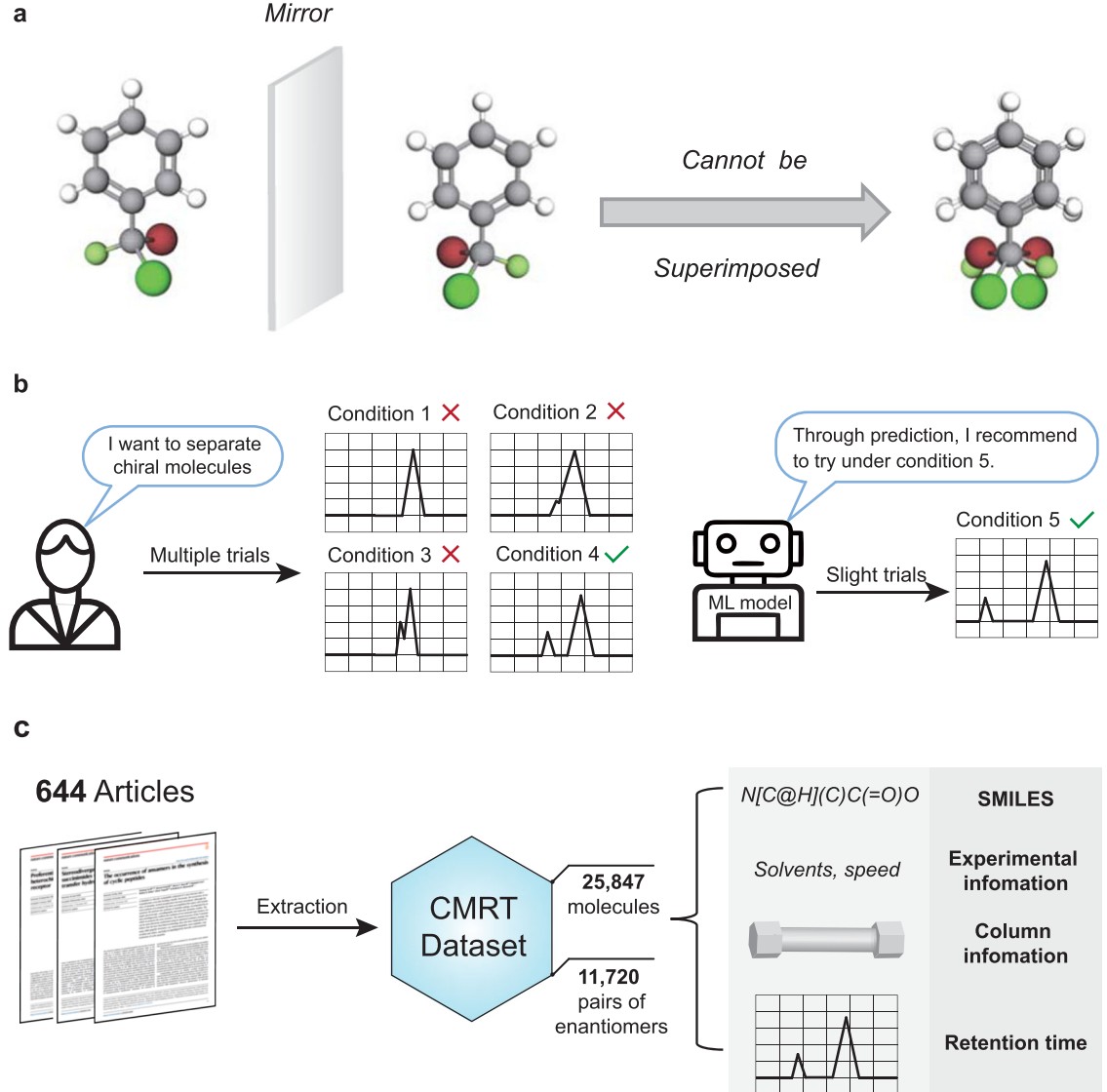

**Fig. 1 | The scheme for chromatographic enantioseparation. a** The diagram for chiral molecules, which are mirror images of each other, but not superimposable. **b** The comparison between classic separation ways by multiple trials with different conditions and machine learning (ML) models that can recommend the most suitable conditions with the highest separation probability. **c** The generation procedure and contents of the chiral molecular retention time dataset (CMRT dataset) in this work. Experimental data of 25,847 molecules (including 11,720 pairs of enantiomers) are extracted from 644 articles about asymmetric catalysis, containing SMILES, experimental information, HPLC column information, and corresponding retention time.

the retention time of 25,847 molecules, which contains 11,720 pairs of enantiomers, experimental information, and HPLC column information. The molecules are recorded in the form of SMILES (Fig. 1c).

2. A machine learning framework called quantile geometry-enhanced graph neural network (QGeoGNN) is constructed by combining quantile learning and GeoGNN, which takes the data uncertainty and chiral molecular representation into consideration and shows satisfactory performance in predicting retention times of chiral molecules. In this framework, the domain knowledge of chromatography and experimental conditions are also incorporated into the model to enhance its extendibility.

3. The prediction model can guide chromatographic enantioseparation by predicting the separation possibility in different conditions and thus eliminating repeated trials, which provides a framework for the utilization of machine learning techniques to facilitate experimental chemistry where data are expensive and unstandardized.

## Results

### Backgrounds and the CMRT dataset

As a ubiquitous phenomenon in nature, molecular chirality is a significant factor that affects molecular properties. A pair of chiral molecules is termed as enantiomers, which are mirror images of each other, but not superimposable (Fig. 1a). Although the molecular constitution of enantiomers is identical with the same atoms and bonds, their properties may be disparate due to the chirality. As an example, left-handed thalidomide is an effective tranquilizer for parturition, while the right-handed enantiomer leads to developmental abnormality in fetuses, and the mixture of enantiomers in the drug once triggered a tragedy[27]. Generally, separating chiral molecules is a challenge because the constitutions of chiral molecules are identical. In order to obtain enantiomerically pure compounds, several chromatographic enantioseparation techniques have been developed in the past decades to separate and analyze the chiral compounds[28]. Among these techniques, high-performance liquid chromatography (HPLC) becomes the mainstream way benefiting from its high efficiency and popularity[21]. In HPLC, the retention time (RT) is a fundamental characteristic, which is defined as the time of chromatographic components from injection to peak (Fig. 1b). It can be used as a qualitative basis for chromatographic enantioseparation since each compound corresponds to a retention time under certain condition. In the chromatographic enantioseparation, the normal-phase HPLC column is adopted where stereoregular chiral polymers are employed as chiral stationary phases (CSPs) to differentiate chiral molecules. Considering that there exist diversified CSPs, different types of HPLC columns have discrepant chiral recognition capacities for different chiral compounds. However, the choice of experimental conditions, including the column type, flow speed, and elution proportion, is currently determined by experience and repeated trials (Fig. 1b). Therefore, this work attempts to construct a machine learning prediction model to predict retention times of chiral molecules and thus facilitating chromatographic enantioseparation.

To achieve this goal, the CMRT dataset is established by automatically extracting experimental results from the relevant literature, the extraction procedure of which is provided in the Method section. Several interesting aspects of the field of asymmetric catalysis can be reflected in the statistical analysis of the dataset including the contribution of authors, the average new enantiomers reported in the literature, and the usage frequency of HPLC columns, which are detailed in Supplementary Information S1.1. Meanwhile, visualization of molecules in the CMRT dataset is provided in Supplementary Fig. 3.

### Construction of QGeoGNN

Benefiting from the natural graphic attribute of molecular structure, graph representation has attracted increasing attention in recent years[29]. The atoms and chemical bonds in the molecule are easy to be interpreted as a graph, which is referred to as Graph G (Fig. 2a). The node and edge features in Graph G are related to the characteristics of the molecular atoms and bonds, respectively. Meanwhile, considering that the bond length and angle can reflect the information of 3D conformation, another bond-angle graph, Graph H, is constructed as a complement for Graph G to incorporate geometry characteristics. In Graph H, the node feature is the bond length and the edge feature is the bond angle (Fig. 2a). Compared with traditional molecular representations, the graph representation can reflect chirality by the chiral tags for labeling the handedness of chiral centers. Based on Graph G and H, the quantile geometry-enhanced graph neural network (QGeoGNN) is constructed. As illustrated in Fig. 2b, the proposed QGeoGNN takes experimental settings (i.e., elution proportion) into consideration, which makes the framework more appropriate to address practical experimental scenes. Meanwhile, the incorporation of relevant molecular descriptors further assists to distinguish enantiomers from macroscopic molecular properties. Through the graph convolution, the graph representations are obtained and then transformed into the prediction through a fully-connected layer. Details for the construction of QGeoGNN are provided in the Method section.

According to the chromatographic process equation[30], there exists an inverse proportional relationship between the retention time and flow rate, which is written as:

$$RT = t_0\left(1 + K\frac{V_s}{V_m}\right) \approx \frac{1}{v}(V_m + KV_s), \tag{1}$$

where $RT$ is the retention time, $K$ is the partition coefficient, $v$ is the flow rate, $V_m$ and $V_s$ are the volume of mobile and stationary phase, $t_0$ is the dead time, respectively. The equation has been verified through experiments in our laboratory with controlled experimental conditions, as described in the Method section. However, since the data used in this study were collected from diverse literature sources, variations in experimental environments may affect the accuracy of the equation. In general, we expect the overall error of the equation to be around 1 (min×mL/min), which is an acceptable level of accuracy for most applications. Upon analysis of the CMRT dataset, we discover that the measured error of the equation is 1.37, which is slightly higher than the expected error but still within an acceptable range (Supplementary Fig. 4). It is important to note that this level of error is not expected to affect the prediction of enantioseparation, as the experimental conditions for enantiomers will be the same. Therefore, the prediction target is set to be $RT×v$ (abbreviated as $RTv$) in this work to incorporate the chromatographic process equation. Another important component of the proposed QGeoGNN is the utilization of quantile learning, which takes uncertainty into account. Conventional retention time prediction tasks usually focused on the accuracy of the predicted retention time while the uncertainty is neglected. However, the experimental error will bring inevitable deviations to the measured retention time. Specifically, in this case, the task of the prediction model is not only to predict the retention time but also to further guide chromatographic enantioseparation. Traditionally, whether enantiomers can be separated is decided by the difference between retention times, and the threshold is very small (usually tens of seconds), which means the uncertainty and errors have a great influence on whether the enantiomers are predicted to be separable. Therefore, this work adopts an alternative way to involve uncertainty to calculate the separation probability. The measurement of uncertainty in deep learning models has been studied extensively and diversified techniques have been proposed, including the Bayesian techniques[31,32], probability

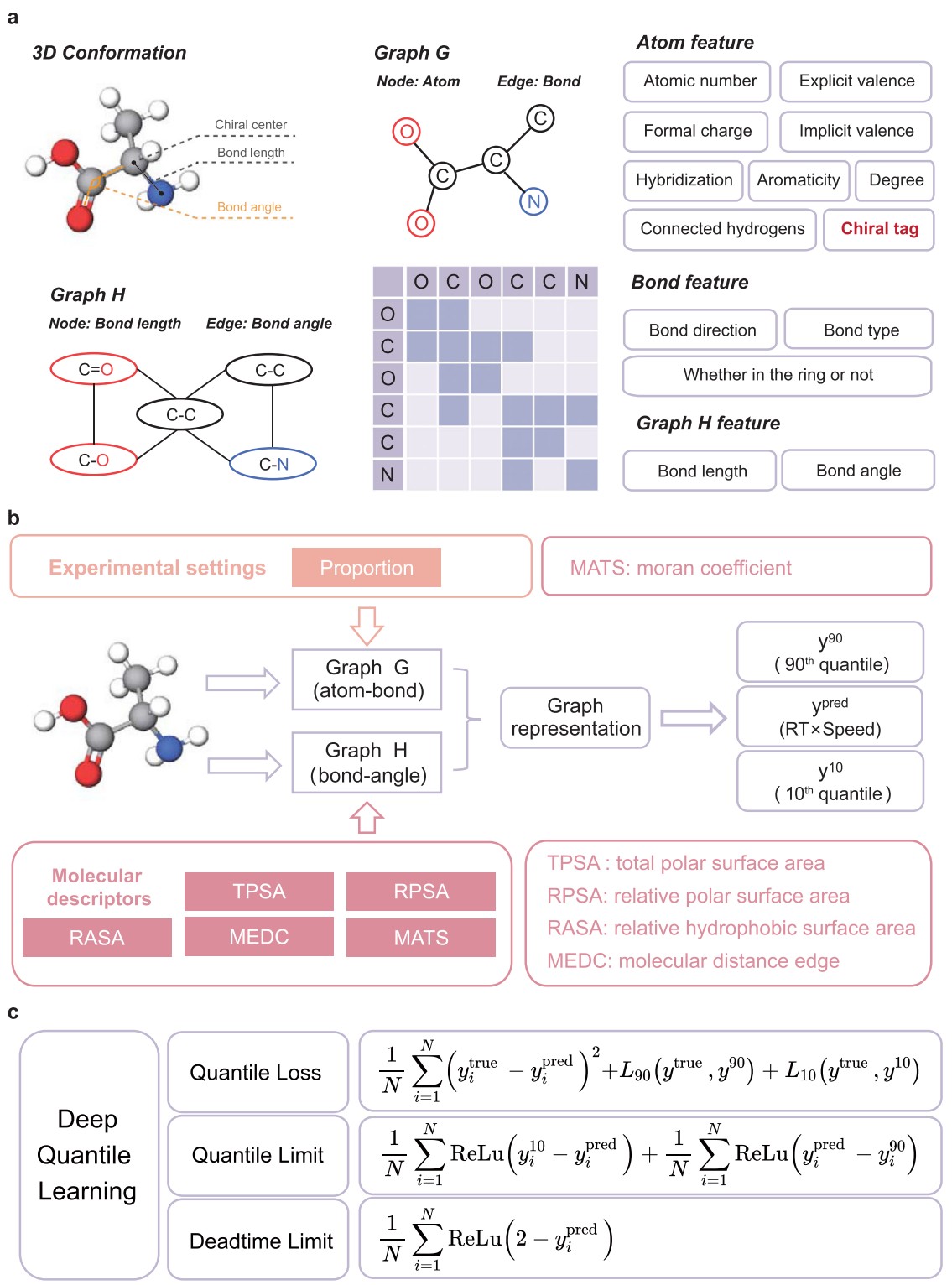

**Fig. 2 | The construction of QGeoGNN. a** An example of the translation from the 3D conformation of a molecule to two graphs including the atom-bond Graph G and the bond-angle Graph H. Each graph consists of node, edge, and corresponding features. The edges are represented in the form of the adjacent matrix. **b** The scheme of the QGeoGNN, the experimental settings, and molecular descriptors are incorporated in Graph G and H, respectively. The output neuron of QGeoGNN is 3, namely, the 90th quantile, the prediction, and the 10th quantile. **c** The constituent of the loss function in QGeoGNN, including the quantile loss, the quantile limit, and the deadtime limit. Here, $y^{\text{true}}$, $y^{\text{pred}}$, $y^{10}$, and $y^{90}$ are the observation, prediction, the predicted 10th quantile, and the predicted 90th quantile, respectively. ReLu refers to the linear rectification function.

distribution modeling[33], and quantile learning[34]. Compared with other methods that require sophisticated modification of the model, quantile learning has better universality and applicability since it can predict the percentiles by simply adding quantile loss to the loss function, which is written as:

$$L_\alpha(y^{true}, y^\alpha) = \sum_{i=y_i^{true} < y_i^\alpha} (0.01\alpha - 1)|y_i^{true} - y_i^\alpha| \\ + \sum_{i=y_i^{true} \geq y_i^\alpha} 0.01\alpha|y_i^{true} - y_i^\alpha| \quad (2)$$

where $L_\alpha$ is the quantile loss, $\alpha$ is the quantile, $y^{true}$ and $y^\alpha$ are the observed data and the quantile prediction. In this work, the loss function of QGeoGNN consists of three parts, namely quantile loss, quantile limit, and deadtime limit (Fig. 2c). The quantile loss enables the QGeoGNN to learn the predicted value, 90th quantile, and 10th quantile simultaneously, while the quantile limit and deadtime limit function as the constraints to make outputs conform to the mathematical and physical restriction.

### Single-column prediction

As illustrated in Fig. 3a, in the area of asymmetric catalysis, various kinds of columns are adopted to handle diversified molecules due to the difficulty of chromatographic enantioseparation. The differences between HPLC column types come from many aspects like CSPs and column model, which affects the chiral recognition ability of HPLC column towards a wide variety of compounds. Among them, ADH, ODH, IC, IA, and OJH are the most frequently utilized column types in the dataset that we collected. From Fig. 3b, it can be seen that the probability density distribution of the retention times in these columns is similar, where the RTs of most molecules are in the range of 5 min to 30 min. To better demonstrate the prediction ability of the proposed QGeoGNN, in this section, single-column prediction is conducted where a prediction model is created in each column type. The benefits of the single-column prediction lie in that the conditions of the column are fixed in each predictive model, which means that the data has a good consistency and is conducive to learning the underlying molecule structure-retention time relationship. For each model, the dataset is split into the training dataset, validating dataset, and testing dataset by 90/5/5. The training dataset is utilized to train the model and the validating dataset is employed for early stopping. The testing data is utilized to examine the model's performance of the out-of-sample prediction. Considering the distribution of RT values, data points with $RTv$ greater than 60 are dropped. The predicted results and corresponding mean average error (MAE), median relative error (MRE), and $R^2$ are shown in Fig. 3c. It is observed that the QGeoGNN has a good predictive ability for each column with $R^2$ all larger than 0.7 and the MAE all below 3, which indicates that the molecular structure-retention time relationship has been learned well.

To eliminate the influence of randomness in splitting the dataset, cross-validation is conducted for each column type, and the results are shown in Supplementary Fig. 5. From cross-validation, it is discovered that the model's performance is restricted to some extent since there exist multiple mild outliers, which means the prediction accuracy of the model for some samples needs to be improved. As intuitively discovered in the previous literature[25], the performance of the ML model usually depends on the similarity between the predicted and trained molecule's structure. Considering that the advantage of using the ML model instead of just using literature search for retention time prediction and chromatographic enantioseparation is to be able to predict the enantiomers not reported in the literature, the generalization ability of the model to enantiomers with different levels of similarity is further investigated. In this study, the Tanimoto similarity coefficient is employed to measure the similarity between 2D structures of two molecules, the range of which is from 0% to 100%. For each model in cross-validation, the Tanimoto similarity coefficient between each enantiomer in the testing dataset and all enantiomers in the training dataset are calculated. Several similarity thresholds are utilized to differentiate the similarity level including 95%, 90%, 80%, 70%, 60%, and 50%. Therefore, all data are categorized into six groups where the molecules in each group as the testing dataset have at least one similar molecule in the training dataset above a specific similarity threshold. The ODH model is taken as an example here, where the size of groups are $n_{95} = 923$ (18.7%), $n_{90} = 1009$ (20.4%), $n_{80} = 1672$ (33.8%), $n_{70} = 3030$ (61.3%), $n_{60} = 3956$ (80.1%), and $n_{50} = 4491$ (90.9%), respectively. Of note, there are 9.1% of enantiomers whose similarity with any molecule in the training dataset do not exceed 50%. The MAE and MRE of the prediction in these groups are illustrated in Fig. 3d. It is confirmed that the generalization ability is highly related to the molecular similarity since the prediction accuracy of the model diminishes evidently with the decrease of similarity. For molecules with over 90% similarity, the model's performance is satisfactory with average relative and absolute errors of only 12.7% and 2.0, while the error raises to 15.7% and 3.0 with over 50% similarity. Compared with existing literature that predicts retention time in UPLC[35] and reversed-phase HPLC[25], the prediction error of our model is slightly higher. It is mainly because the inner similarity of the CMRT dataset obtained from the literature is overall lower than the dataset obtained from actual experiments. Meanwhile, the task of chromatographic enantioseparation in normal-phase HPLC is more complex and the retention times are typically longer.

To better reveal the predictive ability of the proposed QGeoGNN, the influence of data volume and noise is investigated where the prediction on the ODH column is taken as an example. The data noise is added as:

$$\hat{y} = y + \varepsilon \cdot std(y) \cdot N(0,1) \quad (3)$$

where $\hat{y}$ is the noisy data, $y$ is the observation data, $std(y)$ refers to the standard deviation of the observation data, and $N(0,1)$ is the normal distribution. The results are illustrated in Fig. 3d. It is discovered that the QGeoGNN is robust to data noise since the performance maintains stability faced with 10% data noise. At the same time, the observation data itself has inevitable experimental errors, which further verifies the superiority of the proposed method in dealing with noise. In terms of data volume, it is found that the prediction accuracy increases with the increase of training data ratio, and the trend of increase keeps apparent with 90% of the data, which means that if more sufficient data is provided, the prediction accuracy of QGeoGNN still has ample room for improvement.

### Multi-column prediction

On the basis of the satisfactory performance of QGeoGNN in single-column prediction which confirms that the proposed framework is able to learn the molecular structure–retention relationship well, multi-column prediction is conducted in this section that acquires to integrate the prediction of diversified types of columns into a synthetic model. Here, the domain knowledge of chromatography is combined with machine learning techniques to facilitate model construction. In the HPLC column that is depicted in Fig. 4a, CSPs are derived from polysaccharides, including cellulose and amylose which are some of the most common chiral bio-based polymers in nature. As the insufficient chiral recognition ability of cellulose and amylose, their derivatives such as esters and carbamates modified with corresponding substituents are more frequently applied for both analytical and preparative enantioseparations[36]. The CSPs are usually immobilized or coated to silica gel. Therefore, in this work, three main factors are taken into consideration that affect the chiral recognition performance of HPLC columns, including the CSPs, the connection type (immobilized or coated), and the packing material (silica) size.

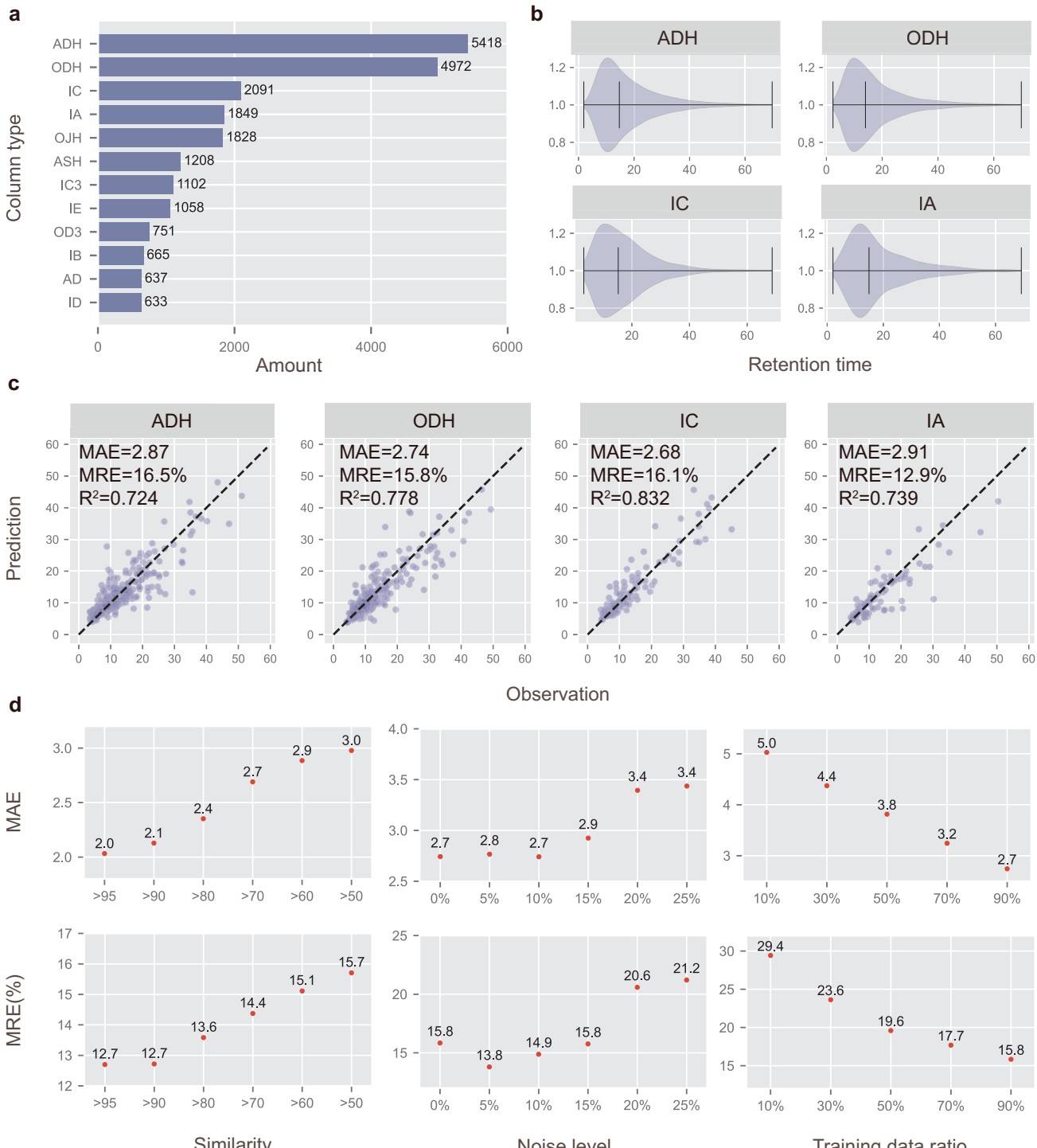

**Fig. 3 | The performance of QGeoGNN for single-column prediction. a** The data amount of each column type in the CMRT dataset established in this work. Only the first 12 column types with the largest volume are displayed here. **b** The violin plot of the retention time in the subset of ADH (5418 data), ODH (4972 data), IC (2091 data), and IA columns (1849 data). The black lines mean the minimum, median, and maximum values, respectively. Molecules with retention time larger than 70 min are rare and are regarded as outliers. **c** Observation versus prediction for the proposed QGeoGNN to predict out-of-sample molecules. Only testing data are shown in plots. The dashed line is the $y = x$ line. The measurements are the mean average error (MAE), median relative error (MRE), and coefficient of determination ($R^2$). **d** The influence of similarity (left), data noise (middle), and data volume (right) on the single-column prediction model. Source data are provided as a Source Data file.

In this work, all types of HPLC columns in the dataset are composed of different collocations of two substrates and seven substituents (Fig. 4a). The substrate is digitized by 0 (amylose) and 1 (cellulose), and the connection type is digitized similarly by 0 (immobilized) and 1 (coated). They are incorporated into the edge features in Graph G of QGeoGNN along with the packing material size (Fig. 4b). The properties of CSPs described by relevant descriptors are added to the edge features in Graph H. As illustrated in Fig. 4b, the edge features of Graph G and Graph H can be represented by feature matrixes and the incorporation of column features can be conducted

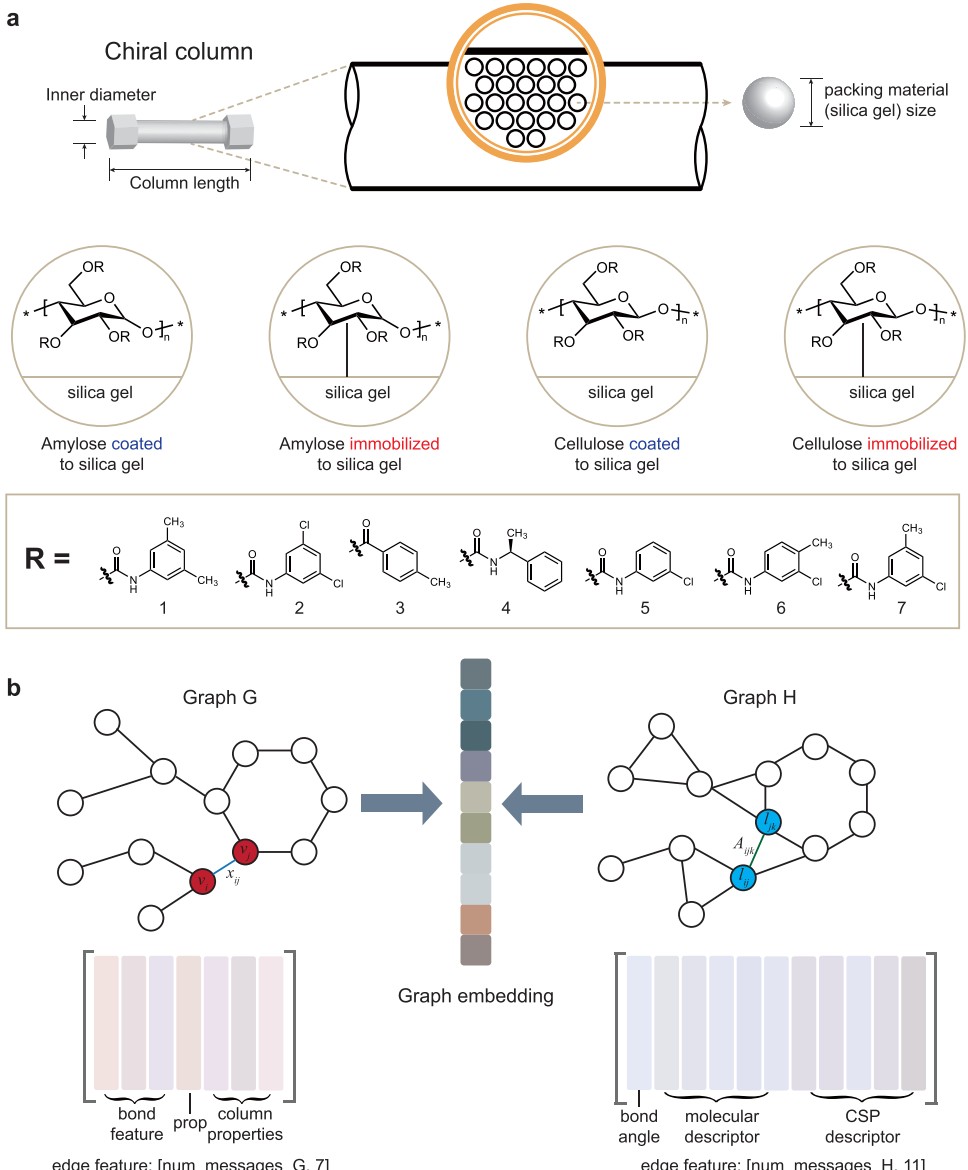

**Fig. 4 | The incorporation of column features in multi-column prediction. a** The domain knowledge of chromatography by HPLC, which is consisted of the chiral stationary phase (CSP) and mobile phase. For CSPs, the packing material size, substrates, substituents, and connection type (immobilized or coated) will affect the chiral recognition ability of HPLC columns. The inner diameter and column length can also influence the chiral recognition ability, but they are kept the same in commercial HPLC columns. **b** The illustration for the incorporation of column information. num_messages_G and num_messages_H refer to the number of message paths (normally equals the number of edges) for Graph G and H, respectively. The column properties include the packing material size, substrates (cellulose or amylose), and connection type (immobilized or coated). The CSP descriptor includes moran coefficient (MATS), total polar surface area (TPSA), relative polar surface area (RPSA), relative hydrophobic surface area (RASA), and molecular distance edge (MEDC) of the CSP.

by the augmentation of the respective feature matrix. Details of the incorporation of column information are provided in the Method section. In this way, all data in the CMRT dataset can be adapted to train a synthetic model for multi-column prediction, which enhances the availability of data. Considering that it is unrealistic to establish single-column prediction models for some less frequently used columns, where the data volume is small and insufficient for model construction, the multi-column prediction models combine the domain knowledge of chromatography with the machine learning model so that it can handle a variety of columns, which further improves the flexibility and scalability of the framework. The predictive performance of the multi-column prediction models is illustrated in Fig. 5a, where the entire data is split into 90/5/5 and the prediction on testing data is depicted in the figure. Faced with data from diversified columns and experimental conditions, the $R^2$ and MAE of the predictive model

still achieve 0.702 and 3.40, which confirms the predictive ability of the synthetic model. An additional experiment is conducted in Supplementary Information S2.3 to observe the influence of column features in the multi-column prediction, and the results show that the incorporation of column information is of great importance for the accuracy of QGeoGNN.

In order to better demonstrate the superiority of the proposed QGeoGNN, conventional machine learning techniques, including LGB, XGB, artificial neural network (ANN), and GNN, are adopted to train prediction models for comparison. In LGB, XGB, and ANN, the molecular fingerprints and descriptors are employed for representation, while GNN only utilizes Graph G for molecular representation. The column information is incorporated into these models as well and other conditions are kept the same. For LGB, XGB, and ANN, the column information is combined with the input fingerprints and

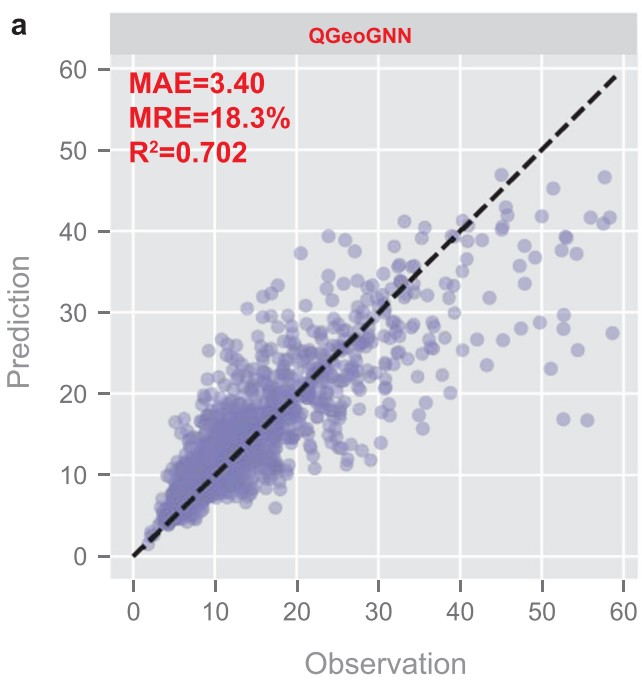

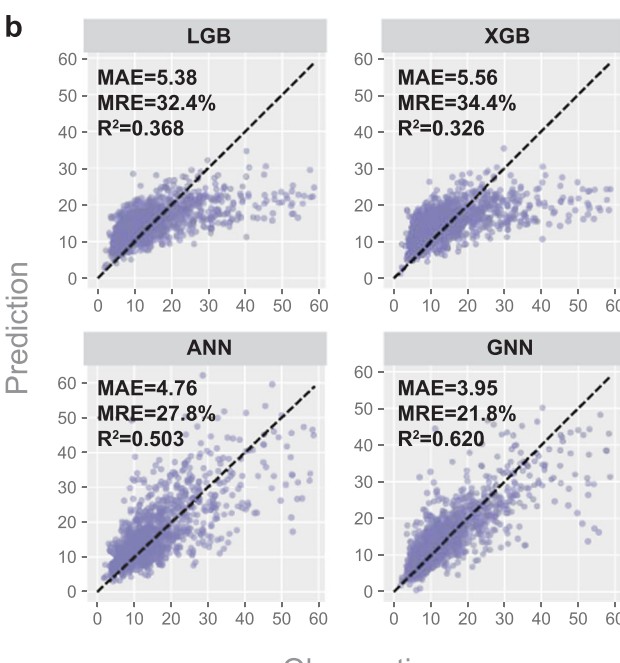

**Fig. 5 | The comparison between QGeoGNN and conventional machine learning techniques for multi-column prediction. a** Observation versus prediction for the proposed QGeoGNN to predict out-of-sample molecules in diversified high-performance liquid chromatography (HPLC) columns. **b** Observation versus prediction for LightGBM (LGB), XGBoost (XGB), artificial neural network (ANN), and graph neural network (GNN) for comparison. Only testing data are shown in plots. The dashed line is the $y = x$ line. The measurements are the mean average error (MAE), median relative error (MRE), and coefficient of determination ($R^2$). Source data are provided as a Source Data file.

descriptors. For GNN, the column information is formed similarly and incorporated in Graph G. More details of implements of these conventional models are provided in the Method section. The results are provided in Fig. 5b. It is obvious that classic tree-based models like LGB and XGB have poor predictive performance. On the other hand, the ANN can grasp the general relationship, but the accuracy needs to be improved. In comparison, the GNN-based models show superior ability where $R^2$ of GNN and QGeoGNN achieve 0.620 and 0.702 while the MAE reaches 3.95 and 3.40, respectively. This is an interesting phenomenon since the conventional tree-based and ANN-based models often perform well in most chemical molecular prediction tasks, including previous literature on RT prediction[24–26]. In order to reveal the reasons behind this phenomenon explicitly, we provide the prediction of two pairs of example enantiomers given by different machine learning models in Table 1. It is evident that the conventional models have difficulty in differentiating enantiomers since the predicted RTs are very close, even the same, which accounts for the poor performance on the CMRT dataset with enantiomers. In contrast, the QGeoGNN can not only distinguish enantiomers well but also provide an accurate predicted RT and its value range. The results demonstrate that the representation of chiral information is of great significance in the chromatographic enantioseparation task considered in this work, and the graph is proven to be a superior representation method than the conventional molecular fingerprints and descriptors when dealing with enantiomers. In addition, compared with GNN, the additional Graph H that incorporates the information of 3D conformation assists to learn the inherent molecular structure–retention relationship, which further improves the predictive ability of the model.

**Chromatographic enantioseparation probability assessment**
The ultimate goal of the retention time prediction model is facilitating chromatographic enantioseparation, which has been an outstanding issue all along. The difficulties for machine learning mainly concentrate on two aspects including chiral representation and error sensitivity. Specifically, chiral representation decides the ability to

distinguish enantiomers while the error sensitivity determines the accuracy. Benefiting from the geometry-enhanced graph representation and the quantile learning, the proposed QGeoGNN provides a promising way to handle the above-mentioned challenges and thus facilitate chromatographic enantioseparation. In order to quantitatively evaluate the probability of enantioseparation under certain experimental conditions like column types, flow rate, and elution proportion, a chromatographic separation probability $S_p$ is proposed based on the predicted value ranges of enantiomers. In this work, $S_p$ is defined based on the naïve principle that the area within the overlapping part of the value range is considered inseparable, while other areas are separable. The conceptual formula can be written as:

$$S_p = \frac{L_{\text{separable}}}{L_{\text{total}}} \quad (4)$$

where $L_{\text{separable}}$ refers to the length of the value range in which the enantiomers are predicted to be separable and $L_{\text{total}}$ refers to the total length of the value range. As detailed in the Method section, the separation probability $S_p$ is derived and calculated as:

$$S_p = 1 - \frac{\max(0, RT_{90}^{\min} - RT_{10}^{\max})}{RT_{90}^{\max} - RT_{10}^{\min}} \quad (5)$$

where $RT_{90}^{\max}$ and $RT_{90}^{\min}$ are the maximum and minimum of 90th percentiles for both enantiomers, $RT_{10}^{\max}$ and $RT_{10}^{\min}$ are the maximum and minimum of 10th percentiles, respectively (Fig. 6a). The unit of $RT$ is minute while $S_p$ is dimensionless. The defined chromatographic separation probability $S_p$ ranges from 0 to 1. A higher $S_p$ refers to a higher possibility that the enantiomers are predicted to be separable by the model. Some examples are given in Fig. 6a to better illustrate the separation probability. It is found that the proposition of $S_p$ assists to eliminate the impact of prediction errors to some extent. Specifically, single predicted values have low fault tolerance since the separation threshold is rigorous (usually tens of seconds). In

**Table 1 | Predicted retention time of two pairs of example enantiomers given by different machine learning models, including LightGBM (LGB), XGBoost (XGB), artificial neural network (ANN), and the proposed QGeoGNN**

|  |  |  |  |  |
|---|---|---|---|---|
| Reference | 20.1 | 23.7 | 17.2 | 19.7 |
| XGB | 15.1 | 15.1 | 17.1 | 17.1 |
| LGB | 14.7 | 14.7 | 17.3 | 17.3 |
| ANN | 33.1 | 30.2 | 14.0 | 14.0 |
| QGeoGNN | 19.3–22.0 (20.7) | 22.8–26.0 (24.5) | 16.8–19.1 (18.0) | 18.9–21.3 (20.1) |

The result of QGeoGNN is presented in the form of the 10th quantile to 90th quantile (predicted retention time)

comparison, quantile learning provides potential value ranges in consideration of uncertainty, which can provide a separation probability instead of simple yes or no, which improves fault tolerance rate, and is more meaningful for chromatographic separation. Meanwhile, chromatographic enantioseparation prediction requires the model to learn the difference between enantiomers well. It means that although the prediction error in multi-column prediction (MRE = 18.3%) seems to be much greater than the separation threshold, chromatographic enantioseparation prediction is still possible.

In order to demonstrate the ability of the proposed model to facilitate chromatographic enantioseparation, several experiments are conducted in this section. First, 412 pairs of enantiomers (i.e., 824 data) are randomly selected from the CMRT dataset to form the testing dataset while the training dataset (23,020 data) and validating dataset (904 data) are randomly chosen from the remaining data to train the prediction model. The separation probability $S_p$ of the enantiomers in the testing dataset is calculated and illustrated in Fig. 6b. Considering that the proposed separation probability is a naïve estimation of the probability of enantioseparation under certain experimental conditions, the threshold to determine whether enantiomers are regarded as separable is decided in the practical scene to be 0.38, which is detailed in Supplementary Information S2.4. Therefore, in this work, we regard those enantiomers with $S_p > 0.38$ as separable, and the accuracy for the separation of 412 pairs of enantiomers reaches 85.7%. The high accuracy confirms the ability of the proposed QGeoGNN to predict chromatographic enantioseparation. In order to assess the influence of molecular similarity on the accuracy of enantioseparation prediction, additional investigations are conducted using the same methodology as described above. Specifically, we use six similarity thresholds (>95%, >90%, >80%, >70%, >60%, and >50%) to group the molecules, resulting in different group sizes: $n_{95} = 4$ (0.97%), $n_{90} = 17$ (4.13%), $n_{80} = 118$ (28.6%), $n_{70} = 273$ (66.3%), $n_{60} = 367$ (89.1%), and $n_{50} = 403$ (97.8%), respectively. As illustrated in Fig. 6c, our results demonstrate that the accuracy of the model is highly dependent on the structural similarity of the enantiomers being predicted. Specifically, the prediction accuracy of the model reaches an accuracy rate of 100% and 94.1% for molecules with a similarity of >95% and >90%, respectively. However, as the similarity threshold decreases, the prediction accuracy of the model also decreases, indicating that the model may have limitations in accurately predicting enantioseparation for more dissimilar compounds. It is worth noting that chromatographic enantioseparation prediction differs from typical classification tasks owing to its distinctive requirements. Considering the difficulty of chromatographic enantioseparation, a suitable separation condition is very important but scarce, which means that it is unaffordable to predict a separable situation as inseparable (Type I error) since it may miss the precious

suitable separative conditions. In contrast, it is relatively acceptable to predict the inseparable condition as separable (Type II error), because this will only induce additional experiments. Benefiting from the multi-column prediction accomplished by QGeoGNN, the separation probabilities of the same enantiomers in different types of HPLC columns can be obtained and compared, which can directly reflect the predicted chiral recognition ability of each column to the given enantiomers, thus providing suggestions on selecting proper experimental conditions without tedious trails and errors. In Fig. 6d, we provide an example of the utilization in practical application. To separate enantiomers, multiple candidate conditions composed of six column types and corresponding proportions and flow rates are considered to select the most proper separative condition. The selected column types are those frequently utilized in chromatographic enantioseparation and commonly seen in the organic laboratory. For each column, the proportion and flow rates of the candidate condition are determined by a domain expert that is likely to generate a suitable retention time. If experiments are conducted to try with all these conditions, it will take several hours. In contrast, the QGeoGNN only needs seconds to predict the separation probability under each condition, which can be visually depicted in the figure, and easy to find the most proper situations with the largest $S_p$ and moderate predicted retention time, thus saving appreciable time for experimenters. The experimental experiments confirm that the enantiomers can only be separated in the IG column, which is consistent with the prediction. For comparison, four conventional techniques including XGB, LGB, ANN, and GNN, are trained under the same condition and tested with the same enantiomer and candidate conditions. The $\triangle RT$ is calculated for each of the candidate conditions, which is denoted as $|RT_1 - RT_2|$. The results are depicted in Fig. 6e. Traditionally, whether the enantiomers can be separated is decided by the $\triangle RT$, and the separation threshold is usually 0.3 min (black dotted line in Fig. 6e). It can be seen that ANN, LGB, and XGB cannot distinguish enantiomers since the predicted retention time of enantiomers is similar and even the same ($\triangle RT$ close to 0). Therefore, these three methods tend to predict all enantiomers to be inseparable, which proves that they do not have the ability for chromatographic enantioseparation prediction. It is found that GNN can learn the difference between enantiomers, however, the prediction is completely wrong. As shown in the figure, it predicts ADH, IC, and ID to be separable while ODH, IG, and ASH are inseparable. Nevertheless, the real situation is that only the IG column can separate the enantiomers. It proves the accuracy of GNN is limited and insufficient to provide correct guidance for practical experiments. The comparison further proves the superiority of the proposed QGeoGNN and separation probability in chromatographic enantioseparation prediction.

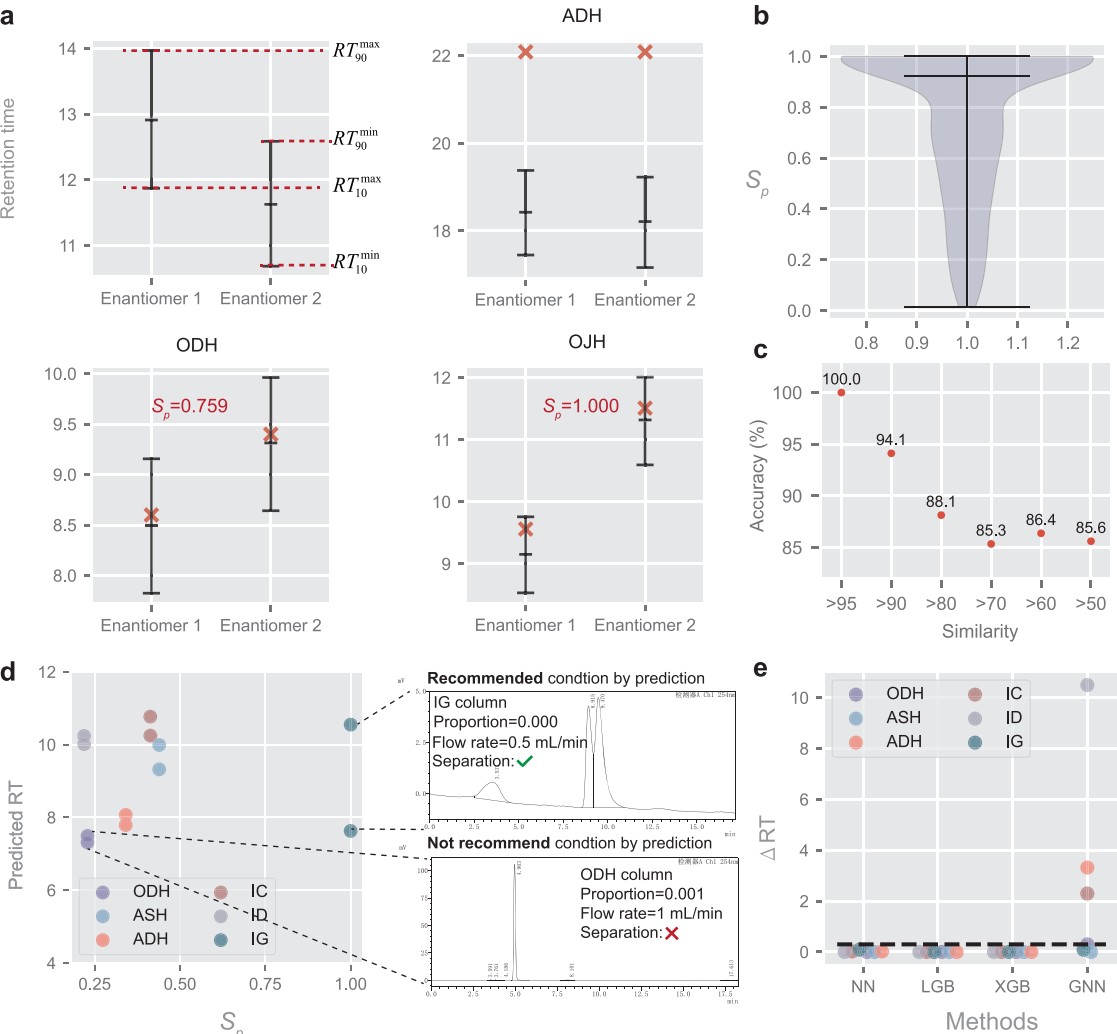

**Fig. 6 | Definition and application of Chromatographic enantioseparation probability assessment. a** The definition of separation probability and some examples under different conditions, including an inseparable situation under the ADH column and two separate situations under ODH and OJH columns. The red cross refers to the observed data while the lines in error bar refers to 90th quantile, predicted retention time (RT), and 10th quantile, respectively, which are predicted by the model. **b** The violin plot of the distribution of calculated $S_p$ for 412 pairs of testing enantiomers. The black lines mean the maximum, median, and minimum values, respectively. **c** The accuracy of enantioseparation prediction for the enantiomers with different thresholds of similarities. **d** An example of the utilization in practical application, including prediction of retention time and $S_p$ for the same enantiomers under different columns made by multi-column prediction model (left), and verification experiments (right). $S_p$ refers to the separation probability. **e** The $\triangle RT$ for each of the candidate conditions with four conventional ML methods. The dark dotted line refers to the separation threshold. Source data are provided as a Source Data file.

## Discussion

In this work, a research framework is proposed to incorporate machine learning techniques into the field of experimental chemistry to promote the efficiency of the researchers practically when faced with chromatographic enantioseparation. The proposed framework of quantile geometry-enhanced graph neural network (QGeoGNN) focuses on addressing several core issues including data acquisition, chiral molecular 3D representations, and data uncertainty. Firstly, Benefiting from the consistency of standardized commercial HPLC columns, this work manages to construct a chiral molecular retention time dataset from numerous articles in the area of asymmetric catalysis and thus solve the problem of data acquisition. Secondly, a specialized graph neural network called QGeoGNN is established by incorporating molecular 3D conformation, experimental conditions, relevant descriptors, and quantile learning to be more suitable for experimental practice. Considering that the experimental results inherently have uncertainty, the quantile learning technique attempts to capture the uncertainty during the training process and can provide a value range. Thirdly, domain knowledge like chromatographic process equation

and HPLC column features is combined with machine learning techniques to further improve the predictive performance of the model. Finally, separation probability is defined in this work to measure the predictive probability of enantiomers being separated under a given condition to facilitate chromatographic enantioseparation.

Experiments have confirmed that the QGeoGNN has a satisfactory ability to predict the retention time of chiral molecules in single-column and multi-column predictions. Furthermore, the QeoGNN can predict the separation probability in diversified conditions quickly and flexibly, and recommend suitable conditions by comparison, which will promote the efficiency of chromatographic enantioseparation.

At present, this research remains some shortcomings that can be improved in the future. Firstly, the representativeness and quality of the data are uncontrolled and sometimes biased since they are extracted from existing literature, which will affect the predictive performance of the machine learning model. Second, due to the lack of repeated test data reported in the literature for the same molecule, data uncertainty is learned from similar molecules. Third, prediction accuracy still needs to be improved when faced with unfamiliar

molecules with low similarity. Lastly, the feature extraction process could be further optimized to better represent chiral-related information. Despite these limitations, we believe that this framework has significant potential to facilitate the experimental process by enabling more efficient and effective determination of proper experimental conditions in chromatographic enantioseparation.

## Methods

### The graph representation for molecules

In this work, the molecular representation is accomplished by constructing two graphs, Graph G and H (Fig. 2a). In data science, graphs are often used to describe unstructured data like social networks, chemical molecules, and traffic networks. A typical graph is composed of several nodes and edges that indicate the connection relationship. As illustrated in Fig. 2a, Graph G expresses the plane structure of the chemical molecule where the nodes refer to the atoms and the edges refer to the chemical bonds. The feature of each node in Graph G contains 9 properties of the corresponding atom, including the atomic number, chiral tag, degree, explicit valence, formal charge, hybridization, implicit valence, aromaticity, and the number of connected hydrogen atoms. The feature of each edge in Graph G contains 3 properties of the corresponding bond including bond direction, bond type, and whether in the ring or not. In this work, the experimental condition, elution proportion, is also added to the feature of each bond. On the other hand, Graph H describes the 3D conformation of the molecule where nodes refer to the bond and edges refer to the bond angle (Fig. 2a). The feature of each node in Graph H contains only one property, the length of the corresponding bond, while the feature of each edge in Graph H contains six properties including the bond angle and five relevant descriptors, namely total polar surface area (TPSA), relative polar surface area (RPSA), relative hydrophobic surface area (RASA), molecular distance edge (MEDC), and moran coefficient (MATS). The molecular descriptors are calculated by the python package *Mordred* and chosen according to the spearman coefficient that identifies the correlation with retention time. In multi-column prediction, the features of HPLC columns are incorporated in the edge feature of Graph G.

### Details for QGeoGNN

The optimization of the graph neural network (GNN) is accomplished based on the message-passing mechanism. Specifically, for node $i$, its representation vector $h_i^k$ at the $k^{th}$ iteration can be written as:

$$a_i^k = A^k(h_i^{k-1}, h_{j \in N(i)}^{k-1}, x_{ij}),$$ (6)

$$h_i^k = C^k(h_i^{k-1}, a_i^k)$$ (7)

Here, $A^k$ and $C^k$ are the aggregation function and the update function in the $k$th iteration. They function as aggregating messages from a node neighborhood and updating the node representation[16]. $N(i)$ is the neighborhood of node $i$, and $x_{ij}$ is the edge that connects node $i$ and its neighborhood node $j$. In the final iteration, the readout function, i.e., the pooling function, is employed to obtain the graph representation $h_G$ from the node representations in the final iteration $K$. The formula can be expressed as:

$$h_{Graph} = R(h_i^K | i \in I)$$ (8)

where $R$ is the readout function and $I$ is the collection of all nodes in the graph. In this work, the readout function is the summation.

The GeoGNN proposed in this work involves the massage passing in two graphs, Graph H and G, simultaneously. Therefore, its optimization mechanism is a little more complex, which is detailed in this section. In GeoGNN, the node representations in Graph H are first calculated in the same way as Eq. (6), which can be formalized as:

$$a_{H_i}^k = A_H{}^k(h_{H_i}^{k-1}, h_{H_j \in N(H_i)}^{k-1}, x_{H_i H_j}),$$ (9)

$$h_{H_i}^k = C_G^k(h_{H_i}^{k-1}, a_{H_i}^k)$$ (10)

where $H_i$ and $H_j$ are the node $i, j$ in Graph H. Considering that the node of Graph H and the edge of Graph G are both related to the bonds of the molecules, a bridge that transforms information between Graph H and G is constructed. Therefore, the node representations in the Graph G are obtained in consideration of the $h_{H_i}^{k-1}$, which is written as:

$$a_{G_i}^k = A_G{}^k(h_{G_i}^{k-1}, h_{G_j \in N(G_i)}^{k-1}, h_{H_i}^{k-1}),$$ (11)

$$h_{G_i}^k = C_G^k(h_{G_i}^{k-1}, a_{G_i}^k)$$ (12)

It can be seen that the information of Graph H is incorporated into the node representations of Graph G. Then, the readout function is conducted to obtain the graph representation of QGeoGNN as:

$$h_{Graph} = R(h_{G_i}^K | i \in I)$$ (13)

In this way, the information of 3D conformation for the molecule including the bond length and bond angle is incorporated into the QGeoGNN to get a graph representation that can distinguish the enantiomers. Considering that the $h_{Graph}$ is a vector with the dimension of embedding size, a fully connected layer is adopted to transform the graph representation into the prediction.

### Quantile geometry-enhanced graph neural network (QGeoGNN)

The QeoGNN is constructed based on the graph representation and the graph isomorphism network (GIN). The fundament of GIN is the graph isomorphic convolution layer (GINConv)[37], which is defined as:

$$x_i' = h_\theta \left( (1 + \varepsilon) \cdot x_i + \sum_{j \in N(i)} x_j \right)$$ (14)

where $x_i'$ and $x_i$ are the node representations in the next layer and current layer, respectively. $x_j$ is the representation in the adjacent nodes. $h_\theta$ is a multilayer perceptron (MLP) and $\varepsilon$ is a constant that equals 0 in this work. In the QGeoGNN, the node embedding is performed for each node in Graph H to obtain the corresponding node representation based on GINConv. Then, the node representation of Graph H is added to the edge representation of Graph G to build a bridge for information transmission between Graph G and Graph H. Afterwards, the node representation of each node in Graph G obtained by node embedding is pooled to get the graph representation. Finally, a fully-connected layer is used to transform the graph representation into the prediction. Deep quantile learning is incorporated into the QGeoGNN by modifying the loss function.

Benefiting from the deep quantile learning technique employed in this work, the quantiles of the prediction that is seen as a variable can be obtained. In this work, we choose the 90th quantile and 10th quantile as the upper and lower bound to measure the uncertainty of the predicted RT$v$. Therefore, the output neuron of the fully-connected layer is 3. The quantile learning is accomplished by the

quantile loss in the loss function, which is written as

$$Loss = \frac{1}{N}\sum_{i=1}^{N}\left(y_i^{\text{true}} - y_i^{\text{pred}}\right)^2 + L_{90}\left(y^{\text{true}}, y^{90}\right) + L_{10}\left(y^{\text{true}}, y^{10}\right)$$
$$+ \frac{1}{N}\sum_{i=1}^{N}\text{ReLu}\left(y_i^{10} - y_i^{\text{pred}}\right) + \frac{1}{N}\sum_{i=1}^{N}\text{ReLu}\left(y_i^{\text{pred}} - y_i^{90}\right) \quad (15)$$
$$+ \frac{1}{N}\sum_{i=1}^{N}\text{ReLu}(2 - y_i^{\text{pred}}).$$

The loss function is composed of three parts including the quantile loss, quantile limit, and deadtime limit. The quantile loss function is defined in Eq. (2). The essence of quantile loss is a segmented function, which separates overestimated and underestimated cases and gives different coefficients. Through the quantile loss, the different quantiles of the target value can be learned during training. It is worth noting that the mean squared error is utilized in predicting the RT$v$ instead of the mean absolute error in conventional quantile learning because we find that it is more suitable for the optimization of the geometry-enhanced graph neural network. Meanwhile, some physical and mathematical constraints, including quantile relationships and the dead time limit. For example, the relationship $y^{10} \leq y^{\text{pred}} \leq y^{90}$ should be satisfied. In the HPLC column, dead time refers to the retention time of components that do not interact with fixation, which is the lower bound of the retention time (RT). In this work, the deadtime limit of RT$v$ is 2, which is decided based on the analysis of the datasets. These constraints in the loss function can avoid the QGeoGNN to make predictions that do not conform to physical and mathematical laws, which improves its accuracy and confidence.

## The incorporation of column information

For multi-column prediction, the column information is incorporated into the QGeoGNN. In this section, the details for the incorporation of column information are detailed. As illustrated in Fig. 4b, QGeoGNN involves two graphs, the atom-bond graph (Graph G) and the bond-angle graph (Graph H). In the graph neural network, the graph structure is described in the form of the adjacent matrix (Fig. 2a). Considering that the adjacent matrix is usually sparse, a compact form called the coordinate matrix (COO matrix) is adapted to represent the sparse matrix to improve storage and computing efficiency. Therefore, the edge index of a graph can be represented in the form of a COO matrix with the size of [2, *num_messages*]. The variable *num_messages* refers to the number of massage paths, which normally equals the number of edges. The feature of each edge corresponds to a relevant vector that incorporates several important attributes. Therefore, the feature for all edges can be expressed by a matrix with the size of [*num_messages*, *num_features*], where the variable of *num_features* refers to the number of relevant attributes. In the multi-column prediction, the connection type, substrates, and packing material size are incorporated into the edge features in Graph G. The properties of CSPs are described by relevant descriptors and are added to the edge features in Graph H. As illustrated in Fig. 4b, the edge feature of Graph G turns to be the size of [*num_messages_G*, 7] by adding the attributes of the column's properties, including the packing material size, substrates (cellulose or amylose), and connection type (immobilized or coated), and each of them is represented by a vector with the size of [*num_messages_G*, 1]. Of note, the substrate is digitized by 0 (amylose) and 1 (cellulose), and the connection type is digitized similarly by 0 (immobilized) and 1 (coated). Similarly, the column descriptors (CSP descriptors) are incorporated into Graph H to augment the feature matrix to the size of [*num_messages_H*, 11]. In

this work, the graph neural network (GNN) with merely graph G is employed for comparison. Therefore, for GNN, all column information including the column's outer features and descriptors are added to the edge feature matrix of Graph G along with the molecular descriptors to generate a matrix with the size of [*num_messages_H*, 17].

## Automatic construction of the CMRT dataset

In this work, a chiral molecular retention time (CMRT) dataset is established by extracting experimental outcomes from 644 articles about asymmetric catalysis. The dataset constitutes the retention time of 25,847 molecules, which contains 11,720 pairs of enantiomers, experimental information, and HPLC column information. The basic flows of constructing the dataset involve several major steps, including determining the data sources, downloading the supplementary information, converting the format, extracting the information, and preprocessing the data. First, 18 research groups that are committed to the research of asymmetric catalysis for years are considered to be the data sources. Then, the supplementary information of relevant articles (644 articles) is downloaded from the journal websites successively and manually. Afterward, the pages of HPLC experimental reports in each article are extracted and converted to text format (.txt). The converted texts from each article are copied and combined into a separate text file. Benefitting from the similar format of reporting the experimental outcomes in the literature of asymmetric catalysis, the experimental results can be extracted automatically through natural language processing techniques, which is detailed below. Finally, the extracted data are pre-processed to obtain the formatted data, where rapid verification is conducted manually to exclude data with obvious errors caused in the extraction process.

Benefitting from the similar format of reporting the experimental outcomes in the literature of asymmetric catalysis, we can extract the data automatically. A typical experimental report in the article is provided in Supplementary Figure. 1 as an example[38]. This experimental report is usually placed in the supplementary materials of the relevant literature, which have been downloaded beforehand. For each reported chiral molecule, there will exist a corresponding experimental report. A program is written to extract data automatically and the principle is detailed below. As shown in Supplementary Fig. 1a, the molecular name is located by the word (S) or (R), which indicates that the compound is a chiral molecule with fixed conformation. After extracting the molecular name, the verification word 'HPLC' is utilized to decide whether this report employs the HPLC to recognize the molecule. If not, the report will be dropped and the subsequent one will be considered. When satisfying the demand of the verification word, all relevant information is extracted step by step including column type, elution proportion, flow rate, the retention time for this conformer, and its enantiomer by corresponding keywords (Supplementary Fig. 1b).

The extracted raw data are saved in *csv* format. After the raw data are obtained, further treatment is conducted to extract the simplified column name from the column type, and convert the proportion to the ratio. For example, the elution proportion in Supplementary Fig. 1 (98/02) will be transformed into 0.02. Considering that the report only includes two retention times (major and minor), we only contain the molecules with less than two chiral centers since the retention times and enantiomers can be matched easily. The extracted names of enantiomers are converted into SMILES through a website (https://cactus.nci.nih.gov/chemical/structure). Specifically, an automatic program written by Python is employed to open the URL that converts molecule names to SMILES. For example, to obtain the SMILEs of D-Lactose, the program will open the URL https://cactus.nci.nih.gov/chemical/structure/D-Lactose/smiles and read the converted SMILES. Other molecule names can be obtained in the same manner by replacing the 'D-Lactose' with the molecule's name in the above URL.

## Experimental verification of chromatographic process equation

In this work, in order to reduce the variable considered in retention time prediction, the chromatographic process equation is considered to construct the relationship between the retention time and the speed, which is written in Eq. (1). Therefore, under a fixed experimental condition where the $K$, $V_m$, and $V_s$ are fixed, the retention time for the molecule and the flow rate is inversely proportional related. Therefore, in this work, the prediction target is set to be RT×$v$, which is abbreviated as RT$v$. In this section, the chromatographic process equation is verified by an experiment which is illustrated in Supplementary Fig. 4. In the verification, the retention times (RT) of a pair of enantiomers with different flow rates $v$ are measured in the laboratory. The experimental condition is the IG column with elution proportion=0.001. In some situations, the outcome is measured repeatedly. The fitted curve and observed retention times are illustrated in Supplementary Fig. 4a. It is discovered that the chromatographic process equation functions well when the flow rate is not so small (bigger than 0.2). Fortunately, there are only smaller than 0.5% data where the flow rate is smaller than 0.2, which means that most data in the dataset satisfies the chromatographic process equation. Therefore, it is rational to incorporate the chromatographic process equation into QGeoGNN to satisfy the underlying relationship between the retention time and the speed and facilitate model construction.

According to the chromatographic process equation and the above experiments in the laboratory, the RT$v$ should keep constant with different flow rates when other experimental conditions are fixed. Nevertheless, considering that data in the CMRT dataset are collected from various literature, errors will emerge in the RT$v$ model because of the difference between the experimental environment of diversified laboratories. Therefore, the scalability of the proposed RT$v$ model in the dataset is further investigated. Here, 364 pairs of the same enantiomer acquired in different flow rates are selected from the CMRT dataset, where the flow rates range from 0.3 mL/min to 1 mL/min and the retention time range from 3.5 min to 60 min. Most of these enantiomers are measured in multiple laboratories independently. In order to measure the error range of the RT$v$ model, the absolute error of the calculated RT$v$ for the same pair of enantiomers in different flow rates is obtained and analyzed. The violin plot of the error distribution is displayed in Supplementary Fig. 4b. The median error is 1.37. It is discovered that there exists a certain degree of error in the RT$v$ model but most of them are small. Meanwhile, the existence of a few samples with extremely large errors implies that there may be a few exceptions to the chromatographic process equation or mistakes in the data.

## Derivation of chromatographic separation probability

In this work, a measurement denoted as chromatographic separation probability $S_p$ is defined to measure the probability that the ML model would correctly separate the enantiomers under the specific experimental setting. In this section, the definition and derivation of chromatographic separation probability are detailed. Benefiting from the proposed QGeoGNN, the value range of the enantiomers' retention time can be obtained. The definition of chromatographic separation probability $S_p$ is defined based on the naïve principle that the area within the overlapping part of the value range is considered inseparable, while other areas are separable. Therefore, the definition of $S_p$ can be written as Eq. (4). In order to derive the formula of $S_p$, two typical scenarios are discussed. The example is illustrated in Supplementary Fig. 7. For the situation in Supplementary Fig. 7a where the value ranges of enantiomers are partially overlapped, the overlapping region is predicted to be inseparable while other regions are seen

as separable. Therefore, in this situation, $L_\text{separable}$ and $L_\text{total}$ can be calculated as

$$L_\text{total} = RT_{90}^\text{max} - RT_{10}^\text{min} \tag{16}$$

$$L_\text{separable} = L_\text{total} - L_\text{inseparable}$$
$$= \left(RT_{90}^\text{max} - RT_{10}^\text{min}\right) - \left(RT_{90}^\text{min} - RT_{10}^\text{max}\right) \tag{17}$$

the $S_p$ can be derived as

$$S_p = \frac{L_\text{separable}}{L_\text{total}} = 1 - \frac{RT_{90}^\text{min} - RT_{10}^\text{max}}{RT_{90}^\text{max} - RT_{10}^\text{min}} \tag{18}$$

Here, $RT_{90}^\text{max}$ and $RT_{90}^\text{min}$ are the maximum and minimum of 90th percentiles for both enantiomers, $RT_{10}^\text{max}$ and $RT_{10}^\text{min}$ are the maximum and minimum of 10th percentiles, respectively. For a special case where the value ranges of enantiomers are completely overlapped, $S_p$ will be equal to 0 since the $L_\text{separable} = 0$. For the situation in Supplementary Fig. 7b where the value ranges of enantiomers are disjoint. It is obvious that the enantiomers are predicted to be separable within the range of error, and the $S_p = 1$. In general, the $S_p$ can be summarized as

$$S_p = \begin{cases} 1 - \frac{RT_{90}^\text{min} - RT_{10}^\text{max}}{RT_{90}^\text{max} - RT_{10}^\text{min}}, & RT_{90}^\text{min} \geq RT_{10}^\text{max} \\ 1, & RT_{90}^\text{min} < RT_{10}^\text{max} \end{cases} \tag{19}$$

which can be simplified as

$$S_p = 1 - \frac{\max(0, RT_{90}^\text{min} - RT_{10}^\text{max})}{RT_{90}^\text{max} - RT_{10}^\text{min}} \tag{20}$$

Therefore, the defined chromatographic separation probability $S_p$ ranges from 0 to 1. A higher $S_p$ refers to a larger region of separable, that is, a higher possibility that the enantiomers are predicted to be separable by the QGeoGNN.

## Experimental settings and parameters

In the QGeoGNN utilized in this work, the number of GINConv is 5, the graph pooling strategy is the summation, the embedding dimension of the node and edge representation is 128, and the batch size is 2048. The training epoch is 1500, and the validate loss is adapted for early stopping. The optimizer is Adam and the learning rate is 0.001. For single-column prediction, the prediction models are established for ADH, ODH, IC, and IA columns, respectively. For each column, the sub-dataset is randomly divided into 90/5/5 to obtain the training, validating, and testing dataset. For multi-column prediction, the entire dataset is split into 90/5/5 to train a synthetic model. For comparison, the XGB, LGB, ANN, and GNN are also employed to train a predictive model. The input of XGB, LGB, and ANN is composed of the 167-dimensional MACCS keys that are utilized to represent the molecular structure, the 5-dimensional molecular descriptors that are the same as those utilized in QGeoGNN, and 3-dimensional column information. For XGB, the number of estimators is 200, the maximum depth is 3, and the learning rate is chosen to be 0.1. For LGB, the maximum depth is 5, the learning rate is 0.007, the number of leaves is 25, and the number of estimators is 1000. For ANN, there are 3 hidden layers with 50 hidden neurons in each hidden layer. The activation function is leaky ReLu and the optimizer is Adam with a learning rate of 0.001. The training epoch is 10,000 and early stopping is adopted. The construction of GNN is similar to QGeoGNN while it only has Graph G, and the loss function is of the mean squared error between the predicted and observed value. The column information is incorporated into the edge features for GNN.

## Reporting summary

Further information on research design is available in the Nature Portfolio Reporting Summary linked to this article.

## Data availability

The CMRT dataset generated in this study have been deposited in the Github repository, https://github.com/woshixuhao/Retention-Time-Prediction-for-Chromatographic-Enantioseparation/tree/main/dataset. Source data are provided with this paper.

## Code availability

All original code has been deposited at the website https://github.com/woshixuhao/Retention-Time-Prediction-for-Chromatographic-Enantioseparation/tree/main/code. The version of the record of the GitHub repo is doi:10.5281/zenodo.7623903.

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

## Acknowledgements

The authors thank Prof. Changkun Li at Shanghai Jiao Tong University for invaluable discussions and helpful suggestions. This work is supported by the Natural Science Foundation of China (Grant Nos. 22071004, 21933001, 22150013, received by F.M.).

## Author contributions

F.M. constructed the chiral molecular retention time (CMRT) dataset. J.L. and H.X. analyzed the data. H.X. performed chemoinformatic and

machine learning studies. H.X. and F.M. wrote the manuscript. F.M. conceived the idea and designed the overall research. F.M. and D.Z. supervised the whole project.

## Competing interests

F.M., H.X., and D.Z. are inventors on a patent application (CN 2023105190181) submitted by Peking University that cover a chiral separation prediction algorithm based on deep learning. J.L. declares no competing interests.
