## [Peer Review File · Nature Communications]

REVIEWER COMMENTS

Reviewer #1 (Remarks to the Author):

Xu et al introduce an RT prediction framework using a new chiral RT dataset consisting of pairs of enantiomers. The paper is interesting, but I have some concerns that need to be addressed before I can make a recommendation about whether this paper should be considered for this journal.

- To account for flow rate and solvent composition, the authors use an RT_v model to make all RTs comparable across papers and methods. This is a subtle modelization. The authors prove that this assumption can be adopted based on a small experiment in their lab. Could the authors demonstrate the scalability of this model using pairs of the same enantiomers acquired in different flow rates and solvents, with data from their CMRT dataset? This would allow the authors to determine the error range of the RT_v calculation model.

- The authors use a low validation/test set size of just 10% in total (5% each). This is a low percentage. I suggest using cross validation with the entire dataset e.g., training the model using a different 5% test set successively, until the entire (100%) dataset is predicted.

- Related to the previous point, the advantage of using ML instead of just using literature search to find the best enantioseparation method is to be able to predict the separation of molecules not reported in the literature. However, ML models suffer from generalization error: ML performs well at predicting the properties for molecules that have been trained with, but its performance worsens for molecules that the model has not been trained with. In this case, it has been shown that the ML performance depends on the similarity between the predicted and trained molecule's structure, as intuitively shown in this paper (10.1038/s41467-019-13680-7, Figure 3). It would be interesting to see a similar analysis for this study, to assess how the performance varies depending on the structural similarity (tested on different similarity thresholds) among structures used for training and prediction. I believe that the generalization error/performance should be a central theme in this paper and not just a small discussion at the end of the paper.

- The CMRT dataset automatic construction is quite intriguing, but not enough details are provided. Specifically, details about how the data is mined, how are the supp. materials downloaded, from what journals, etc, are missing. How do the authors retrieve the unequivocal molecular structure (inchi/smiles) from just the name/synonym without errors? This is not described.

- The chromatographic enantioseparation probability (formula #4) is not scholarly described. It is clear from Fig 5 but not from the text, and the units for each of the variables are missing. Also, how is the formula derived? For me, a 90% probability would imply e.g., that there is a 90% of probability that the model would correctly separate the molecules using the specific column type. Based on the boxplots from figure 5 (a, ODH panel), an 80% probability implies a 50% of probability that the molecules will be separated. In addition, it is not described how the separation thresholds: "low (<0.2), medium (0.4-0.6)..." are defined, and they seem randomly picked. The authors could calculate these thresholds more precisely using real data: check whether the probability formula can predict whether the enantiomers can be separated or not, and compare this prediction with the real data. Overall, I suggest the authors clarify the details of the probability framework.

Minor:

- Please, use relative errors (mean/medium errors in seconds) to report model performance.

- Supp. Materials, Section 2.1, reads: "a precious chiral molecular retention time (CMRT) dataset is established...". A precious? The authors should bear in mind that they are writing a scientific paper, not a Tolkien novel ;)

Reviewer #2 (Remarks to the Author):

The authors use a quantile learning based graph neural network to predict retention times in enantioseparation using HPLC.

There are quite a few issues with the manuscript, of which a significant one is the lack of necessary detail to evaluate it properly. I do not support publication in its current form.

Quantile regression is added to loss function of the graph neural network, but the argument for using it is somewhat unconvincing. The authors simultaneously claim that the use of quantile learning in deep learning is innovative (it has been studied somewhat extensively in the recent past in the literature), that it can be used to predict the various quantiles of the distribution of the output, and that the quantile limit and deadtime limit function are constraints to make outputs conform to mathematical and physical restrictions. Their primary claim seems to be that quantile regression can be used to account for uncertainty; however, specifically for their case, while the experimental data used for training has

uncertainty and errors, it is unclear to me that quantile regression is better than any other method of predicting, for example, the variance of the outputs.

Importantly, how the various HPLC descriptors related to CSP and mobile phase are incorporated into the QGeoGNN for multi-column prediction is unclear. For that matter, how the column information is incorporated into the edge features of the GNN used as one of the comparison methods is unclear.

Another claim is that the models can guide experiments, and its practical manifestation is shown in Figure 5d. Multiple candidate conditions are chosen, and the QGeoGNN is used to predict the separation probability for each of these and choose conditions appropriate for the separation, after which experimental validation was performed. How was this set of candidate conditions chosen? Details are lacking in the manuscript. Also, it is clear that the other models (XGB, LGB, ANN, GNN) can also make predictions in a similar amount of time, though perhaps with slightly lower accuracy). However, an important question is if those models would also recommend the same conditions for separation as the QGeoGNN does in figure 5d.

The writing is poor and there are many grammatical issues and issues either with spelling or the ambiguous use of words/phrases (e.g., 'charity' instead of 'chirality' on page 2, 'floatability of RT' on page 5, 'one-hot code' instead of 'one-hot encoding' on page 9).

We would like to sincerely express our deep gratitude to the editors and referees for their insightful comments and suggestions on the manuscript, which are fair, encouraging, and constructive. After carefully studying the comments, we have made corresponding changes in the revised manuscript. The specific responses to the comments are listed below. In the revised manuscript, the text in **red** is the revision for Referee #1. Text in **blue** is the revision for Referee #2.

Response to Referee #1

1. Xu et al introduce an RT prediction framework using a new chiral RT dataset consisting of pairs of enantiomers. The paper is interesting, but I have some concerns that need to be addressed before I can make a recommendation about whether this paper should be considered for this journal.

Reply: Thanks for your constructive comments that help our work to be more convincing, and your interest in our work! In the revised manuscript, we have added substantial experiments, explanations, and details according to the suggestions. The point-by-point responses are attached below.

2. To account for flow rate and solvent composition, the authors use an RT_v model to make all RTs comparable across papers and methods. This is an astute modelization. The authors prove that this assumption can be adopted based on a small experiment in their lab. Could the authors demonstrate the scalability of this model using pairs of the same enantiomers acquired in different flow rates and solvents, with data from their CMRT dataset? This would allow the authors to determine the error range of the RT_v calculation model.

Reply: Thank you for your important and insightful comments. In the Supplementary Information S3.1, the experiments prove that the RT_v model is reasonable and accurate in the laboratory (Fig. S4a). Considering that data in the CMRT dataset are collected from various literature, errors will emerge in the RT_v model because of the difference between the experimental environment of diversified laboratories. As suggested by the referee, we conduct an additional experiment to examine the scalability of the RT_v model and calculate the error range of this model. In the revised Supplementary Information S3.1, we add an experiment where 364 pairs of the same enantiomer acquired in different flow rates are selected from the CMRT dataset to calculate the error of RT_v model. The error range is calculated and illustrated in the form of the violin plot (Fig. S4b). The median error of RT_v model is 1.37. It is discovered that there exists a certain degree of error in the RT_v model but most of them are small. Considering that data in the CMRT dataset are collected in various literature, the error is acceptable. Meanwhile, the existence of a few samples with extremely large errors implies that there may be a few exceptions to the chromatographic process equation or mistakes in the data.

3. The authors use a low validation/ test size of just 10% in total (5% each). This is a low percentage. I suggest using cross-validation with the entire dataset e.g., training

the model using a different 5% test set successively, until the entire (100%) dataset is predicted.

Reply: It is good advice to use cross-validation to further evaluate our proposed prediction model. Therefore, we add the experiments of cross-validation in the revised manuscript (page 7) and Supplementary Information S3.2. As suggested by the referee, the model is trained using a different 5% testing dataset successively until the entire dataset is tested. Therefore, 20 models with different testing datasets are trained independently. The results are shown in Fig. S5. From cross-validation, it is discovered that the model's performance is overall satisfactory but is still affected by multiple mild outliers, which means the prediction accuracy of the model for some samples needs to be improved. This phenomenon intrigues further study on the generalization ability of the proposed model on different similarity thresholds in the revised manuscript (page 7, paragraph 2), which is also pointed out by the referee.

4. Related to the previous point, the advantage of using ML instead of just using a literature search to find the best enantioseparation method is to be able to predict the separation of molecules not reported in the literature. However, ML models suffer from generalization error: ML performs well at predicting the properties for molecules that have been trained with, but its performance worsens for molecules that the model has not been trained with. In this case, it has been shown that the ML performance depends on the similarity between the predicted and trained molecule's structure, as intuitively shown in this paper (10.1038/s41467-019-13680-7, Figure 3). It would be interesting to see a similar analysis for this study, to assess how the performance varies depending on the structural similarity (tested on different similarity thresholds) among structures used for training and prediction. I believe that the generalization error/performance should be a central theme in this paper and not just a small discussion at the end of the paper.

Reply: We are very grateful for the highly valuable comments, which help our work to be completer and more convincible. As noted in the previous reply (Reply 3), it is discovered that the prediction model performs worse on some molecules from cross-validation. Therefore, it is necessary to investigate the relationship between the model's performance and molecular similarity to further reveal the generation ability of the prediction model. We have carefully read the paper by Domingo-Almenara, X. *et al.*, which provides a thorough study on the performance of the retention time prediction model and molecular structure's similarity, and conducted a similar analysis in the revised manuscript (page 7, paragraph 2). Similarly, the Tanimoto similarity coefficient is employed to measure the similarity between 2D structures of two molecules, and different similarity threshold is defined to form different groups. Here, the analysis is conducted based on cross-validation, so all molecules in the dataset can be covered. In this work, several similarity thresholds are utilized to differentiate the similarity level including 95%, 90%, 80%, 70%, 60%, and 50%, and the size of groups are $n_{95}=923$ (18.7%), $n_{90}=1,009$ (20.4%), $n_{80}=1,672$ (33.8%), $n_{70}=3,030$ (61.3%), $n_{60}=3,956$ (80.1%), and $n_{50}=4,491$ (90.9%), respectively. The results of the performance with different similarity thresholds are illustrated (revised Fig. 3d) and discussed (page 7,

paragraph 2) in the revised manuscript. It is confirmed that the generalization ability is highly related to molecular similarity since the prediction accuracy of the model diminishes evidently with the decrease of similarity. This result also explains the slightly higher error of our proposed model than existing literature that predicts retention time in UHPLC and reversed-phase HPLC since the inner similarity of the CMRT dataset obtained from literature is overall lower than the dataset obtained from actual experiments (9.1% enantiomers that the similarity with any molecule in the training dataset do not exceed 50%). Meanwhile, the task of chromatographic enantioseparation in normal-phase HPLC is more complex and the retention times are typically longer. Additional discussion is provided in the revised manuscript (page 8, lines 2~7).

5. The CMRT dataset automatic construction is quite intriguing, but not enough details are provided. Specifically, details about how the data is mined, how are the supp. materials downloaded, from what journals, etc, are missing. How do the authors retrieve the unequivocal molecular structure (inchi/smiles) from just the name/synonym without errors? This is not described.

Reply: Thanks for reminding us about this issue. In the revised manuscript, we add sufficient details about the construction of the CMRT dataset (page 18, Method section and Supplementary Information S2.1). The basic flows of constructing the dataset have several major steps, including determining the data sources, downloading the supplementary information, converting the format, extracting the information, and pre-processing the data (page 18, Method section). For better understanding and further usage, the code for constructing the CMRT dataset is uploaded on GitHub in the open resource. The data source is determined according to the research groups which are committed to the research of asymmetric catalysis for years and are familiar to us. The articles and supplementary information are downloaded from the journal websites successively and manually (page 18, Method section). The extracted names of enantiomers are converted into the SMILES through a website (<https://cactus.nci.nih.gov/chemical/structure>). Specifically, an automatic program written by Python is employed to open the URL and convert molecule names to SMILES automatically (Supplementary Information S2.1). For example, in order to obtain the SMILES of D-Lactose, the program will open the URL <https://cactus.nci.nih.gov/chemical/structure/D-Lactose/smiles> and read the converted SMILES. Other molecule names can be obtained in the same manner by replacing the 'D-Lactose' with the molecule's name in the above URL. The relevant code is also uploaded onto GitHub.

6. The chromatographic enantioseparation probability (formula #4) is not scholarly described. It is clear from Fig 5 but not from the text, and the units for each of the variables are missing. Also, how is the formula derived? For me, a 90% probability would imply e.g., that there is a 90% of probability that the model would correctly separate the molecules using the specific column type. Based on the boxplots from figure 5 (a, ODH panel), an 80% probability implies a 50% of probability that the

molecules will be separated. In addition, it is not described how the separation thresholds: "low (<0.2), medium (0.4-0.6)..." are defined, and they seem randomly picked. The authors could calculate these thresholds more precisely using real data: check whether the probability formula can predict whether the enantiomers can be separated or not, and compare this prediction with the real data. Overall, I suggest the authors clarify the details of the probability framework.

Reply: Thanks for pointing out this issue to us. In the revised manuscript, we supplement substantial details of the probability framework as suggested (page 13, page 14, Supplementary Information S3.5 and S3.6). We have rewritten the formula of chromatographic enantioseparation probability scholarly in the mathematical form (revised Eq. (4) and (5)). The units of each variable are also provided. The detailed definition and derivation of the separation probability are provided in the revised Supplementary Information S3.5. The chromatographic separation probability S_p is defined on the basis of the naïve principle that the area within the overlapping part of the value range is considered inseparable, while other areas are separable. A diagram is illustrated in Fig. S7 for better understanding. The defined separation probability S_p ranges from 0 to 1. A higher S_p refers to a larger region of separable, that is, a higher possibility that the model would correctly separate the enantiomers (page 13). Of note, the proposed separation probability is a naïve measurement, and there may exist other measurements. According to our definition, the ODH panel in the revised Fig. 6a (original Fig.5a) has a 75.9% probability to be separable. The referee seems to regard the probability as $1 - \frac{RT_{90}^{min} - RT_{10}^{max}}{RT_{90}^{min} - RT_{10}^{min}}$ (calculated to be nearly 50% for ODH panel), which can also reflect the separation probability to some extent. However, we have discovered that the S_p defined in this work is closer to the actual situation since the enantiomers that can be separated evidently usually corresponds to a large S_p .

Fig. S7. Examples for calculating separation probability S_p in two typical scenarios

Meanwhile, the separation threshold is of great importance for the enantioseparation prediction. As pointed out by the referee, the initially defined threshold is mechanically and arbitrarily. It is good advice to calculate these thresholds more precisely using real data. Therefore, we precisely calculate the separation threshold to be 0.38 (i.e., 38%) by the real data in the revised manuscript (Supplementary Information S3.6). As suggested by the referee, we compare the prediction with different separation thresholds with the situation of real data and select

the best threshold that achieves the largest synthetic accuracy of enantioseparation. The description of the choice of separation threshold is also revised (page 14).

7. *Please, use relative errors (mean/medium errors in seconds) to report model performance.*

Reply: For comparable reports of results, we use the median relative errors (%) and mean absolute error (min) to report model performance in the revised manuscript and SI as suggested (revised Fig. 3, Fig. 5, Fig. S5, and relevant texts). Of note, the retention time of normal-phase HPLC is typically longer than reverse-phase HPLC and UHPLC since it usually takes tens of minutes. Therefore, we adopt minute (i.e., min) as the unit of the mean absolute error.

8. *Supp. Materials, Section 2.1, reads: "a precious chiral molecular retention time (CMRT) dataset is established...". A precious? The authors should bear in mind that they are writing a scientific paper, not a Tolkien novel ;)*

Reply: Thanks for your kind and humorous notice. We revised such words to be more objective and abandoned the Gollum's pet phrase :)

Response to Referee #2

1. *The authors use a quantile learning based graph neural network to predict retention times in enantioseparation using HPLC. There are quite a few issues with the manuscript, of which a significant one is the lack of necessary detail to evaluate it properly. I do not support publication in its current form.*

Reply: Thanks very much for pointing out the deficiency in our work, which encourages us to make our work complete. In the revised manuscript, we have supplemented substantial details on the issues suggested by the referee to help the readers better understand the construction and contribution of this work. The point-by-point responses are attached below.

2. *Quantile regression is added to loss function of the graph neural network, but the argument for using it is somewhat unconvincing. The authors simultaneously claim that the use of quantile learning in deep learning is innovative (it has been studied somewhat extensively in the recent past in the literature), that it can be used to predict the various quantiles of the distribution of the output, and that the quantile limit and deadtime limit function are constraints to make outputs conform to mathematical and physical restrictions. Their primary claim seems to be that quantile regression can be used to account for uncertainty; however, specifically for their case, while the experimental data used for training has uncertainty and errors, it is unclear to me that quantile regression is better than any other method of predicting, for example, the variance of the outputs.*

Reply: Thanks for the highly valuable comments, which remind us that the explanation for the utilization of quantile learning is insufficient. In the revised manuscript (page 5), we have reviewed ways to measure uncertainty and attached additional arguments

for the usage of quantile learning in this study to be more convincing. We have explained it from two aspects: 1. Why uncertainty measurement is important in this work? 2. Why was quantile learning chosen to measure uncertainty in this work?

As to the first aspect, the consideration of uncertainty is an important component of the framework, which is decided by the attributes of the task. Conventional retention time prediction tasks usually focused on the accuracy of the predicted retention time while the uncertainty is neglected. However, the experimental error will bring inevitable deviations to the measured retention time. Specifically, in this case, the task of the prediction model is not only to predict the retention time but also to further guide chromatographic enantioseparation. Conventionally, whether enantiomers can be separated is decided by the difference between retention time, and the threshold is only about 0.3 min, which means the uncertainty and errors have a great influence on whether the enantiomers are predicted to be separable. Therefore, an alternative way is adopted to measure the separation probability through the predicted value range in this work, which takes uncertainty into account and eliminates the influence of the prediction error on the chromatographic enantioseparation.

On the basis of the first aspect, we further answer the second question that why quantile learning was adopted in this work. As demonstrated above, the measurement of uncertainty is important for chromatographic enantioseparation prediction. As pointed out by the referee, the measurement of uncertainty in deep learning models has been studied extensively and diversified techniques have been proposed, including the Bayesian techniques, probability distribution modeling, quantile learning, and so on. We have reviewed these techniques in the revised manuscript and further analyze their applicability in this work. Most of the above-mentioned techniques require modifying either the model structure or the inputs and outputs. For example, to learn the variance of the outputs, we may adopt the dropout technique and estimate the variance from outputs or parameterize the distribution of the prediction and output the mean and variance through the neural network. However, it will take extra effort to successfully use these techniques on the geometry-enhanced graph neural network (GeoGNN), since its structure and inputs (graphic data) differs from conventional models like ANN or tree-based model. In contrast, quantile learning can predict the percentiles by merely modifying the loss function, which can be directly employed in the GeoGNN model without extra effort. We believe other uncertainty measurement techniques may also function well to evaluate the data uncertainty, but quantile learning has better universality and applicability that can be incorporated into the GeoGNN with the least effort. Therefore, we choose to use quantile learning here. Of note, the main contribution of this work is the consideration of uncertainty to calculate the separation probability to solve the problem of chromatographic enantioseparation prediction. Therefore, we choose quantile learning not because its performance is superior to others, instead, its easy manifestation.

We hope the revised claim will help the referee and potential readers to better understand the usage of quantile learning in this study and the contribution of this work.

3. Importantly, how the various HPLC descriptors related to CSP and mobile phase are

incorporated into the QGeoGNN for multi-column prediction is unclear. For that matter, how the column information is incorporated into the edge features of the GNN used as one of the comparison methods is unclear.

Reply: Thanks for reminding us about this issue. In the revised manuscript (page 11 and Supplementary Information S1.2), the incorporation of column information in the proposed QGeoGNN and the GNN for comparison is demonstrated in detail. In the revised manuscript, we have revised the Fig. 4 to add an illustration for the form of edge features in the QGeoGNN for multi-column prediction for better understanding.

Revised Fig. 4. The incorporation of column features in multi-column prediction.

4. Another claim is that the models can guide experiments, and its practical manifestation is shown in Figure 5d. Multiple candidate conditions are chosen, and the QGeoGNN is used to predict the separation probability for each of these and choose conditions appropriate for the separation, after which experimental validation was

performed. How was this set of candidate conditions chosen? Details are lacking in the manuscript. Also, it is clear that the other models (XGB, LGB, ANN, GNN) can also make predictions in a similar amount of time, though perhaps with slightly lower accuracy). However, an important question is if those models would also recommend the same conditions for separation as the QGeoGNN does in figure 5d.

Reply: We are grateful for the referee to point out this crucial issue. We completely agree that it is very important to investigate whether other models can make a recommendation correctly. Therefore, in the revised manuscript (pages 14~15), we have added an experiment for comparison and provided more details and discussions. For the choice of candidate conditions, we have added some explanations (page 14). Considering that the column type mainly influences whether the enantiomers can be separated, six frequently utilized column types are chosen. For each column, the proportion and flow rates of the candidate condition are determined by a domain expert that is likely to generate a suitable retention time.

As to the comparison between QGeoGNN and other models, we have conducted an additional experiment with the same enantiomer and candidate conditions in the manuscript (pages 14~15). The results are displayed in revised Fig. 6. It is discovered that ANN, LGB, and XGB cannot distinguish enantiomers and tend to predict all enantiomers to be inseparable, which proves that they do not have the ability for chromatographic enantioseparation prediction. Meanwhile, it is found that GNN can learn the difference between enantiomers, however, the prediction of enantioseparation is completely different from the real situation. It proves the accuracy of GNN is limited and insufficient to provide correct guidance for practical experiments. From the comparison, it is confirmed that only QGeoGNN can give the correct recommendation while other methods cannot. We have added some discussion about this issue in the revised manuscript (pages 14~15).

Part of the revised Fig. 6. c, An example of the utilization in practical application, including the enantiomers (left), prediction of different columns made by multi-column prediction model (middle), and verification experiments (right). **d,** The ΔRT for each of the candidate conditions with four ML methods. The dark dotted line refers to the separation threshold.

5. The writing is poor and there are many grammatical issues and issues either with spelling or the ambiguous use of words/phrases (e.g., 'charity' instead of 'chirality' on

page 2, 'floatability of RT' on page 5, 'one-hot code' instead of 'one-hot encoding' on page 9).

Reply: Thanks for pointing out this to us. In the revised manuscript, we have revised the ambiguous words and phrases to keep the paper scholarly. Meanwhile, we also proof-reading the whole manuscript to eliminate grammatical issues and improve the writing.

REVIEWER COMMENTS

Reviewer #1 (Remarks to the Author):

I thank the authors for their thorough revision. But my concerns have been partially addressed. I'll reply using the same comment number as in the response to the reviewers' letter.

2. (Minor) Pg 5, lines 152, please, briefly define quantitatively (and optionally discuss) the expected error range, instead of just stating that it falls under an acceptable error range.

3. The authors have considered the structural similarity for the generalization error, but only for the RT prediction. However, I think that this generalization should be considered in the enantiomer separation prediction since the aim of this work is to correctly predict the separation, not the actual RT. Specifically, page 16, line 393, where the authors state "the accuracy for the 393 separation of 412 pairs of enantiomers reaches 85.7%", is where this accuracy (85.7%) should be calculated based on the structural similarity. The authors could calculate, per each molecule, whether the model can correctly predict whether it can be separated or not, and calculate its similarity to the training set. Then, you can report the actual enantiomer separation accuracy based on the similarity.

6. While the main manuscript is well written, the authors have not given the same attention to the supplementary methods, which form the intellectual part of the work. Therefore, I have not been able to understand the response to point no. 6, where authors use the synthetic recall rate (a term that I have not seen before, and it is not defined or referenced. Do authors mean accuracy?), and terms like TP, FP, TN and FN are not defined. I am not referring to the acronym definition but to e.g., TP includes all pairs of correctly separated enantiomers, TN are...

Also, it is not explained why the S_r value is relatively high almost for the entire range of S_p threshold (Fig S8). The description and discussion of this should also be revised. Finally, why authors have not considered using the established ROC curve to find the best threshold?

Reviewer #2 (Remarks to the Author):

My concerns from the first draft have been addressed satisfactorily, and this is a much improved manuscript that I can support for publication.

We would like to sincerely express our deep gratitude to the editors and referees for their insightful comments and suggestions on the manuscript, which are fair, encouraging, and constructive. After carefully studying the comments, we have made corresponding changes in the revised manuscript. The specific responses to the comments are listed below. In the revised manuscript, the text in red is the revision for Referee #1.

Response to Referee #1

1. I thank the authors for their thorough revision. But my concerns have been partially addressed. I'll reply using the same comment number as in the response to the reviewers' letter.

Reply: Thanks very much for providing further comments, which point out the problems we haven't noticed in the previously revised version. In this version, we have provided experiments, discussions, and more explicit clarifications. The point-by-point responses are attached below.

2. (Minor) Pg 5, lines 152, please, briefly define quantitatively (and optionally discuss) the expected error range, instead of just stating that it falls under an acceptable error range.

Reply: Thank you for your important and insightful comments. As suggested by the referee, we define quantitatively the expected error range and provide more sufficient discussions about the error and its influence on the prediction model (page 5). Considering that the data used in this study were collected from diverse literature sources, variations in experimental environments may affect the accuracy of the equation. In general, we expect the overall error of the equation to be around 1 (min×mL/min), which is an acceptable level of accuracy for most applications. Upon analysis of the experimental data, we discover that the measured error of the equation is 1.37, which is slightly higher than the expected error but still within an acceptable range. It is important to note that this level of error is not expected to affect the prediction of enantioseparation, as the experimental conditions for pairs of enantiomers will be the same.

3. The authors have considered the structural similarity for the generalization error, but only for the RT prediction. However, I think that this generalization should be considered in the enantiomer separation prediction since the aim of this work is to correctly predict the separation, not the actual RT. Specifically, page 16, line 393, where the authors state "the accuracy for the 393 separation of 412 pairs of enantiomers reaches 85.7%", is where this accuracy (85.7%) should be calculated based on the structural similarity. The authors could calculate, per each molecule, whether the model can correctly predict whether it can be separated or not, and calculate its similarity to the training set. Then, you can report the actual enantiomer separation accuracy based on the similarity.

Reply: Thanks for reminding us about this issue. We fully agree with the comments of the referee that it is necessary to assess the influence of molecular similarity on the

accuracy of enantioseparation prediction since the enantioseparation prediction is a core issue of our work. Therefore, we added some experiments to assess the influence of molecular similarity on the accuracy of enantioseparation prediction (page 15) and the results are depicted in the revised Fig. 6c (we add a subplot to show the result). Similarly, we use six similarity thresholds (>95%, >90%, >80%, >70%, >60%, and >50%) to group the molecules and calculate the prediction accuracy of these groups with different similarity thresholds. As illustrated in the revised Fig. 6c, the accuracy of the model is highly dependent on the structural similarity of the enantiomers being predicted. Specifically, the prediction accuracy of the model reaches an accuracy rate of 100% and 94.1% for molecules with a similarity of >95% and >90%, respectively. However, as the similarity threshold decreases, the prediction accuracy of the model also decreases, indicating that the model may have limitations in accurately predicting enantioseparation for more dissimilar compounds.

Fig. 6c. The accuracy of enantioseparation prediction for the enantiomers with different thresholds of similarities.

4. While the main manuscript is well written, the authors have not given the same attention to the supplementary methods, which form the intellectual part of the work. Therefore, I have not been able to understand the response to point no. 6, where authors use the synthetic recall rate (a term that I have not seen before, and it is not defined or referenced. Do authors mean accuracy?), and terms like TP, FP, TN and FN are not defined. I am not referring to the acronym definition but to e.g., TP includes all pairs of correctly separated enantiomers, TN are...

Also, it is not explained why the Sr value is relatively high almost for the entire range of Sp threshold (Fig S8). The description and discussion of this should also be revised. Finally, why authors have not considered using the established ROC curve to find the best threshold?

Reply: Thanks for pointing out this issue to us. Section S3.6 in the SI was written ill-advised, which may lead to some misunderstandings and confusion. Therefore, we have rewritten Section S3.6 to include more clarifications and explanations to make readers easier to understand our intention (pages 13~15 in SI). Instead of the utilization of confusing terms in the prior version, we adopt an explicit definition to describe the objective function in the section. Here, we essentially want to maximize the overall accuracy of both separable and inseparable enantiomers. The new definition is provided in the revised Eq. (S15) on page 14 of SI as:

$$\begin{aligned}
P &= P_{separable} + P_{inseparable} \\
&= \frac{TP}{N_{separable}} + \frac{TN}{N_{inseparable}}, \tag{S.15}
\end{aligned}$$

where $P_{separable}$ and $P_{inseparable}$ are the precision of predicting separable and inseparable enantiomers; TP (True Positive) refers to the number of separable enantiomers to be predicted correctly as separable while TN (True Negative) refers to the number of inseparable enantiomers to be predicted correctly as inseparable. $N_{separable}$ and $N_{inseparable}$ are the numbers of separable enantiomers and inseparable enantiomers, respectively. The revised definition is more clear and easier to be understood, which is similar to the essence of the ROC curve. The results with different thresholds are provided in the revised Fig. S9.

Fig. S9. Overall accuracy with different S_p thresholds.

In the revised manuscript, we also discussed why the overall accuracy is relatively high almost for the entire range of S_p threshold (revised Fig S9). This is because it takes the accuracy of both separable and inseparable enantiomers into account. Specifically, when the threshold is set relatively high, the $P_{separable}$ will decrease and $P_{inseparable}$ will increase, which makes the overall precision maintains high.

We also thank for the referee to provide an alternative way (ROC curve) to find the best threshold. In the revised version, we have added discussion about the ROC curves (page 13) and provides the results of ROC cruve in the revised Fig. S8 for demonstration and comparsion. In fact, we have previously considered the ROC curve, and the essence of the objective function utilized in this section is similar to the ROC curve. The ROC curve essentially identifies the threshold value that balances the trade-off between the accuracy of the positive and negative classes by TPR and FPR. The point that is closest to the top-left corner of the ROC curve is usually regarded as the most proper threshold. However, it is worth noting that the ROC curve requires a sufficient number of samples to generate a smooth curve. If the sample size is relatively small, the ROC curve may be presented in polylines instead of smoothed curves, as shown in the revised Fig. S8. In such cases, the low resolution for thresholds may pose limitations to the interpretation of the curve. Therefore, we have explored alternative

methods for selecting the best threshold, where an objective function (Eq.S15) is established inspired by the essence of the ROC curve, which can explicitly show the overall accuracy under different thresholds with high resolution. Similar to the ROC curve, the overall accuracy defined in this section balances the accuracy of the positive and negative classes. Of note, the best threshold found by the ROC is 0.39, which is close to the 0.38 found in this section, which also proves the effectiveness of the defined overall accuracy.

Fig. S8. The ROC curve for the model to predict the enantioseparation of real data.

REVIEWERS' COMMENTS

Reviewer #1 (Remarks to the Author):

I thank the authors for their responses. All my concerns have been addressed.